# Positional programs in early murine facial development and their role in human facial shape variability

Andrea P. Murillo-Rincón [1,3], Louk W. G. Seton [1,3], Elio Escamilla-Vega [1], Amor Damatac II [1], Janina Fuß [2], Carsten Fortmann-Grote[1] & Markéta Kaucká [1] ✉

The face is a complex, variable structure shaped by environmental and functional adaptations. In humans, the remarkable diversity of facial shapes underpins identity and mutual recognition. The developmental process from cranial neural crest cell migration to facial prominence fusion is conserved and essential for determining facial shape. However, the molecular and cellular underpinnings are not fully understood. We reconstruct facial development in the mouse model at the single-cell level, and show that the facial mesenchyme exhibits a remarkable molecular heterogeneity predominantly driven by positional programs. We then explore the role of these spatially defined murine mesenchymal populations in the extraordinary diversity of human facial shapes. By integrating molecular and spatial coordinates with human genome-wide association studies and genes linked to abnormal human facial shapes, we link genetic variants associated with facial features to individual cell populations and transcriptional signatures. This integrative approach provides a framework for exploring evolutionary processes behind facial variation and offers new insights into congenital facial syndromes.

The face is a remarkably complex structure that exhibits extensive morphological diversity, both intra- and interspecific, across vertebrates. The spectrum of vertebrate facial shapes reflects the adaptation of each species to different environments[1]. In humans, the facial shape also serves as the basis for mutual recognition, assisting social interactions[2]. The face and its shape emerge during embryogenesis in a morphogenetic process orchestrated by conserved genes and developmental programs[3]. Heterochronic, heterometric, and heterotopic changes in the gene expression programs underlie vertebrate morphological diversity and evolution[4,5]. However, the molecular and cellular basis of this modulation remains to be elucidated. Even more intriguing is how intraspecific facial shape variability is generated, particularly in humans, where individual facial features can be inherited in a composite manner from both parents.

In vertebrates, a hallmark of the head and face formation is the specification and migration of the cranial neural crest cells (CNCCs), multipotent progenitors that give rise to the majority of the face[3,6]. In the mouse model, CNCCs are specified within the borders of the closing neural tube, delaminate and initiate migration at embryonic day (E) 8.5. During this time, CNCCs undergo a series of binary decisions[7]. First, the sensory lineage separates from the common autonomic-mesenchymal progenitors, followed by the second stable split resulting in the commitment of cells to the autonomic or mesenchymal lineage. Cells of the mesenchymal lineage subsequently give rise to multiple derivatives and build structures essential for facial morphology, such as cartilage, bone, and soft connective tissue. In mouse, the chondrocranium, the cartilaginous template of the future bony skull, appears around E14.5[8,9]. However, its cellular predecessor, the mesenchymal condensations, can be observed as early as E12.5[10,11].

[1]Max Planck Institute for Evolutionary Biology, Plön, Germany. [2]Institute of Clinical Molecular Biology, Kiel University and University Medical Center Schleswig-Holstein, Kiel, Germany. [3]These authors contributed equally: Andrea P. Murillo-Rincón, Louk W. G. Seton. ✉e-mail: kaucka@evolbio.mpg.de

The mesenchymal condensations and the chondrocranium represent the first geometrical blueprint of the head and the face[12].

This developmental window, spanning from the emergence of the CNCCs to the origin of the skull blueprint, is marked by an extraordinary dynamic sequence of events. Specifically, the mesenchymal cells undergo extensive proliferation to form a substantial cell mass, the main building blocks of facial morphogenesis[13,14]. The mesenchyme and the overlying ectoderm give rise to the facial prominences: mandibular, maxillary, medial nasal and lateral nasal (Fig. 1A). The

prominences grow extensively and fuse to form a normal and functional face[15]. While this process is conserved across amniotes, differences in growth patterns and morphology of the facial prominences are observed across species and are recognized as the source of their characteristic facial proportions[13,16,17]. Despite the importance of this developmental period for face formation, the cellular behavior and underlying molecular programs operating in embryonic space and time remain poorly understood, hindering efforts to uncover the foundations of facial shape variability.

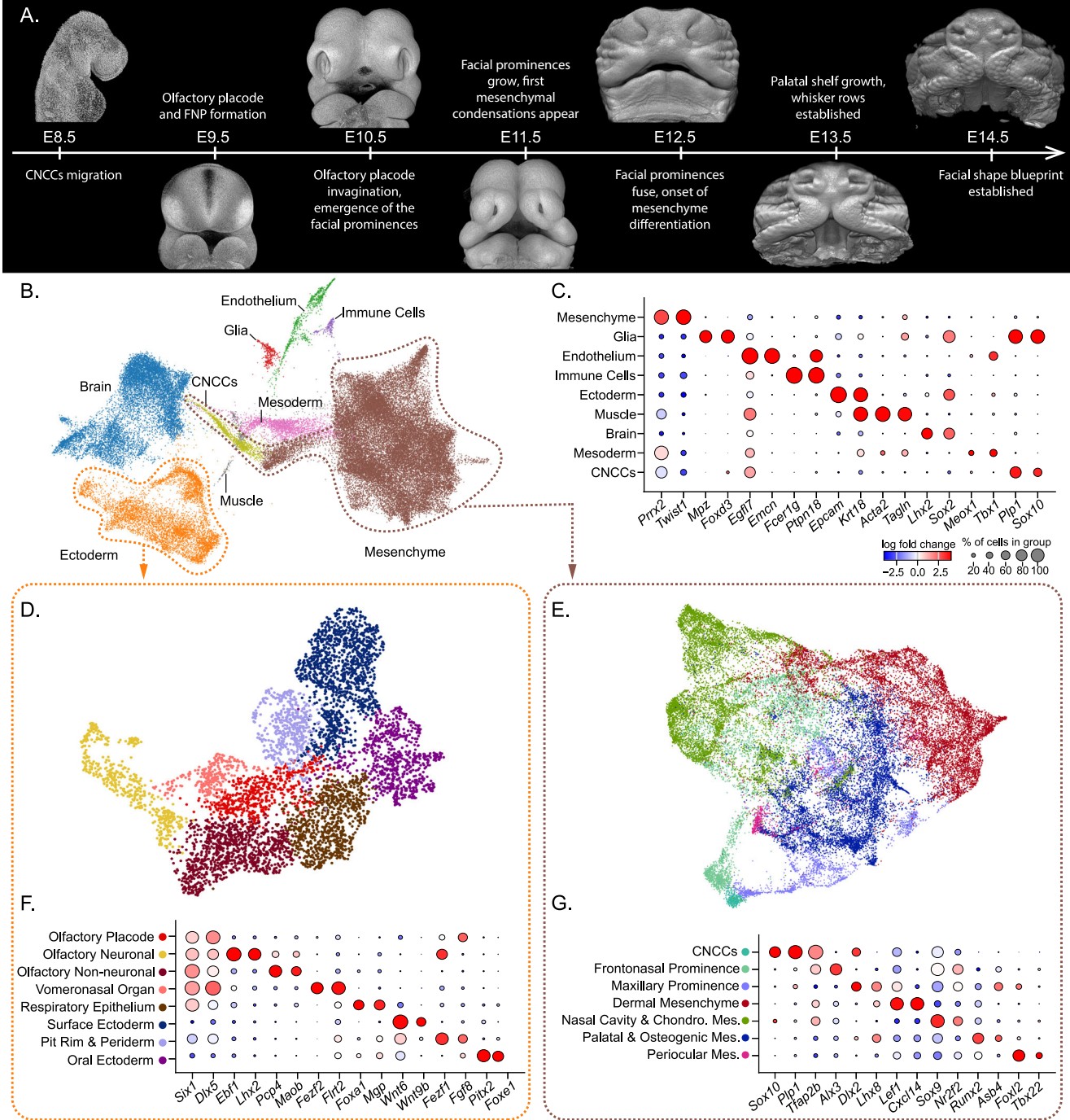

**Fig. 1 | Mouse face development at single-cell resolution. A** Mouse development timeline, from E8.5 to E14.5 (not in scale), highlighting the key events during facial morphogenesis. **B** UMAP visualization of all cells forming the embryonic face between E8.5–E14.5, resulting from the integration of La Mano et al. (2021) (E8.5–E10.5) and our original data (E10.5–E14.5). **C** Dot plot with marker genes used for main cell type identification in (**B**). **D**, **E** Integrated UMAPs of face ectoderm (**D**), and cranial neural crest cells (CNCCs) and mesenchymal (**E**) clusters subset from (**B**), showing the major cell populations composing the developing face. **F**, **G** Dot plots with marker genes used for cluster annotation of mesenchymal (**F**) and ectodermal (**G**) cells.

Here, we set out to reveal the spatiotemporal sequence of cellular and molecular events occurring during this critical period at the single-cell level, with a particular focus on facial mesenchyme, the cellular source of the facial skeleton. Ultimately, using the reconstruction of the murine face developmental trajectory, we seek to identify the molecular programs and cellular sources generating the spectrum of different shapes found in humans. This study will serve as an information-rich reference for evolutionary comparative studies uncovering the basis of intra- and interspecific facial variability across species, as well as a resource providing insights into human congenital facial syndromes.

## Results

### Cellular and molecular trajectories of the developing face in the mouse

To understand the cellular composition and the underlying molecular landscapes governing the morphogenesis of the developing face, we microdissected the upper face area of mouse embryos from five consecutive developmental stages, spanning E10.5–E14.5 (Fig. 1A) and performed single-cell RNA-sequencing (scRNAseq) following the 10X Genomics pipeline. To fully capture the temporal continuum of face morphogenesis, the newly generated dataset was integrated with a previously published dataset from mouse E8.5–E10.5 embryonic head[18]. These two datasets (E8.5–E10.5 and E10.5–E14.5) were generated using animals with CD1 and C57BL/6 N genetic backgrounds, respectively. We applied an integration strategy that corrected for potential batch effects (See Methods; Supp. Fig. 1). Upon stringent quality checks (Supp. Fig. 1; Supp. Table 1), the final dataset contained 58,973 cells representing all expected major cell types, including CNCCs, facial mesenchyme, ectoderm, developing brain, glia, endothelium, mesoderm, muscle, and immune cells (Fig. 1B, C).

In order to resolve the molecular composition of the mesenchyme in greater detail, we subset the mesenchyme and CNCCs clusters and analyzed them together (Fig. 1E, G). We also processed the ectoderm population separately (Fig. 1D, F) to assess cellular communication profiles among ectodermal and mesenchymal populations (see below). We observed that in early stages (E8.5–E11.5), the cluster identity of the mesenchyme was defined by the location of the cells in the embryonic face, i.e., facial prominences, while at later stages (from E12.5 onwards), the clusters mainly reflected the signatures of emerging cell types (Fig. 1E). To better understand the distinct molecular profiles and their transitions, we analyzed the composition of the mesenchyme and the CNCCs cluster at a higher resolution and uncovered 53 mesenchymal clusters representing the differentiation of the CNCCs into facial mesenchyme and its derivatives (Fig. 2A, B).

We performed pseudotime analysis to reconstruct the temporal ordering of cells along development and predicted their entropy, which estimates the differentiation potential of individual cells. We identified seven developmental trajectories corresponding to the dermal whisker pad, osteogenic and chondrogenic lineages, mesenchyme of the nasal cavity, philtrum, palatal shelves and periocular mesenchyme (Fig. 2B). These clusters were annotated using canonical cell type marker genes.

Notably, during early developmental stages (E8.5–E11.5), spanning the formation and growth of facial prominences, the mesenchyme exhibits high entropy, indicating that the cells are not yet committed to any particular differentiation trajectory (Fig. 2D). At these stages, the majority of the cells are in the G2/M or S phase of the cell cycle, indicating overall high proliferation rates of the entire facial mesenchyme (Fig. 2E, F; Supp. Fig. 1). In contrast, at E12.5, the proportion of dividing cells is reduced by half, which, together with a decreasing entropy, indicates the onset of mesenchymal commitment to specific differentiation trajectories. (Fig. 2D; Supp. Fig. 1H). Moreover, fate probabilities also support the onset of cell fate commitment around E12.5 (Fig. 2G).

Interestingly, although the mesenchyme remains largely uncommitted prior to E12.5, RNA velocity analysis showed a molecular split in the FNP mesenchyme as early as E9.5 (Supp. Fig. 2A, B), coinciding with the emergence of the frontonasal (FNP) and maxillary (MxP) prominences. The observed early molecular heterogeneity is consistent with our previous observation that the molecular profiles characterizing the mesenchyme before E12.5 are associated with the position of the cells within the developing face rather than with their differentiation potential. Altogether, our analysis revealed the overall cell behavior dynamics during face development. Early stages are characterized by extensive proliferation and absence of fate commitment, whereas from E12.5 onwards, the facial mesenchyme exhibits decreased proliferation rates and molecular signs of differentiation. These molecular patterns are consistent with the known morphological events taking place in this developmental period (Fig. 1A), such as the extensive outgrowth of facial prominences by E11.5, their subsequent fusion by E12.5 and the appearance of the first mesenchymal condensations, the precursors of the skeletal elements of the face.

### Heterogeneity of the early facial mesenchyme relies on strong expression of positional programs

In order to understand the basis of the molecular segregation present in the early facial mesenchyme suggested by RNA velocity, we analyzed the mesenchyme at high resolution at each individual stage, calculated differentially expressed genes and unique marker genes for each of the resulting clusters. This information formed the basis for extensive cluster validation using whole-mount multiplexed HCR imaging, allowing us to map the position of the clusters in embryonic space and time. The resulting clusters were named based on their position in the developing face, following conventional anatomical terminology (Supp. Fig. 3): proximal (towards or closest to the trunk or point of origin, equivalent to "posterior"), distal (away or furthest from the trunk or point of origin, equivalent to "anterior"), posterior (behind, in the back), dorsal (top), ventral (bottom), medial (close to the midline), lateral (to the sides).

Specifically, at E8.5, the presumptive face is represented by three clusters, corresponding to migrating CNCCs (*Sox10, Plp1*), early frontonasal (FNP; *Tfap2b, Alx3*) and early maxillary (MxP; *Dlx1/2*) mesenchyme (Fig. 3A; Supp. Fig. 4A–C). Remarkably, at E9.5, before the emergence of the facial prominences, the FNP mesenchyme is molecularly segregated into lateral nasal (LNP) and medial nasal (MNP) territories, defined by *Pax7* and *Alx3/Shox2/Gata2* expression domains, respectively[19,20] (Fig. 3B; Supp. Fig. 4D–G). At this stage, the emerging MxP is labeled by the expression of *Dlx2/Meis2/Lhx8* (Fig. 3B; Supp. 4I).

At E10.5, when the facial prominences emerge (Fig. 1A), the molecular complexity of the mesenchyme increases considerably (Fig. 3C). The FNP-derived processes, the LNP (*Pax7*) and the MNP (*Alx3, Pou3f3*), separate both molecularly and spatially, as shown by the expression of early positional genes (Supp. Fig. 5). The dorsal domains of both the LNP and MNP are characterized by the expression of *Tfap2b*. In addition, the MNP exhibits further segregation into medial (*Foxd1/Shox2*) and ventral (*Flrt2/Gata2/Pax9*) regions (Fig. 3C; Supp. Fig. 5). This molecular distinction markedly correlates with the future anterior-to-posterior position of the MNP derivatives in the facial midline (i.e., nasal septum and primary palate). At this stage, the MxP is represented by four partially overlapping clusters that are located at the distal (*Lhx8*), proximal (*Dlx2*), dorsal (*Foxl2, Irx5*) and medial (*Mecom, Meis2*) regions of these prominences (Fig. 3C; Supp. Fig. 5). Interestingly, the medial MxP cluster, located towards the oral cavity, already expresses palatal shelf markers such as *Asb4* and *Barx1*[19,20], suggesting that this population and its location correspond to the cellular source and position of the future palatal shelves. Similarly, the dorsal MxP (*Foxl2, Irx5*) shows expression of *Lef1*, a conventional

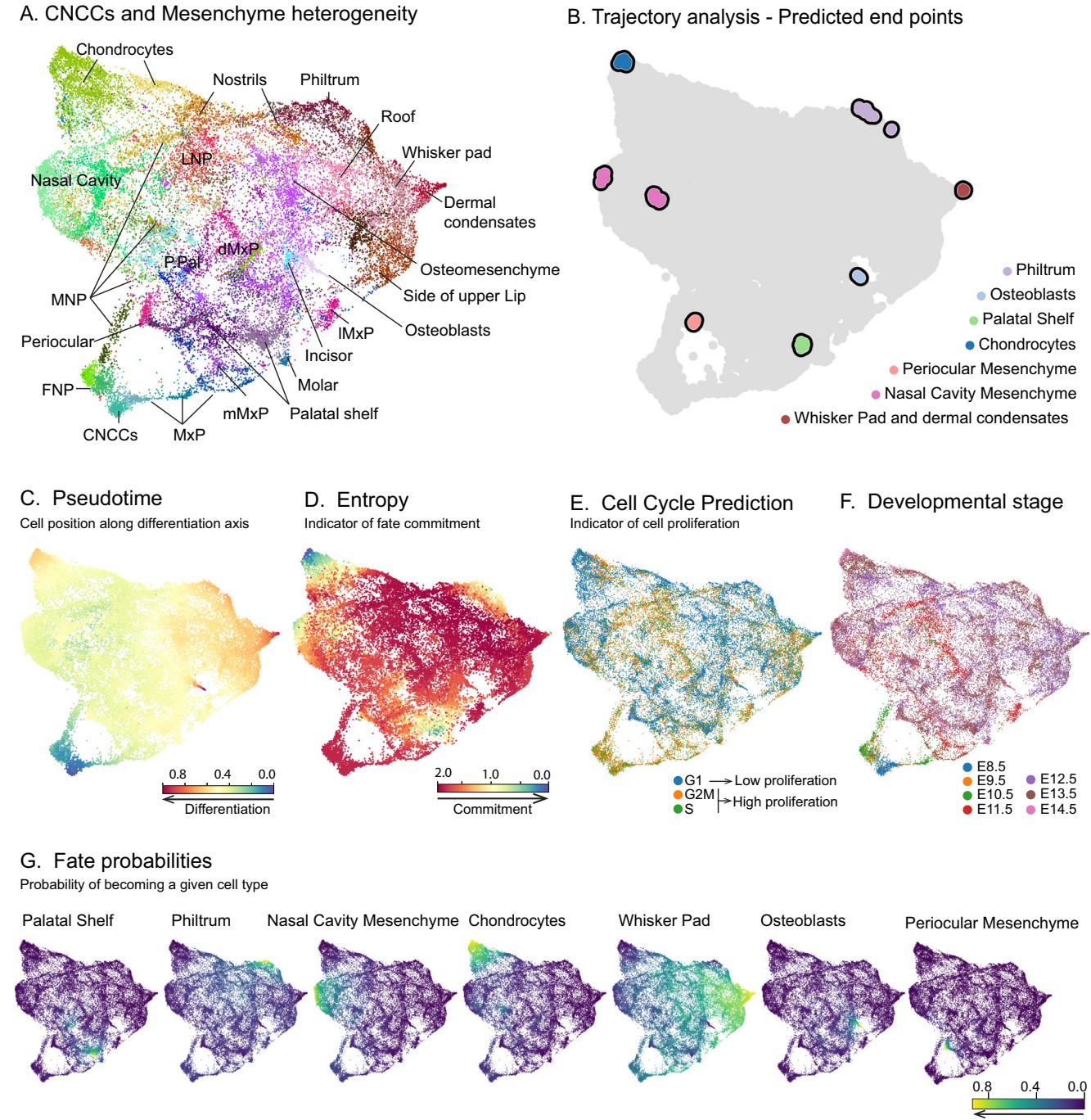

**Fig. 2 | Molecular heterogeneity and developmental trajectories of the facial mesenchyme from E8.5 to E14.5. A** Integrated UMAP visualization of the 53 clusters representing cranial neural crest cells (CNCCs) and facial mesenchyme, analyzed at high resolution. Only representative clusters are named. **B** UMAP visualization of the seven endpoints recovered by the trajectory analysis. **C** Pseudotime and **D** Entropy analyses show the predicted cell differentiation status of the developing facial mesenchyme. High pseudotime and low entropy values indicate higher cell commitment and differentiation probabilities along the developmental trajectories. **E** UMAP visualization for cell cycle prediction and **F** developmental stages of CNCCs and mesenchymal cells employed in this analysis. **G** Fate probabilities for each of the seven trajectories visualized on the integrated UMAP. A higher value indicates a higher probability that a cell acquires a given fate. FNP frontonasal prominence, LNP lateral nasal prominence, MxP maxillary prominence, mMxP medial maxillary prominence, dMxP dorsal maxillary prominence, lMxP lateral maxillary prominence, MNP medial nasal prominence, P. Pal primary palate.

marker of the developing dermis[21], posterolaterally, in the proximity of the overlying ectoderm, anticipating the emergence of the whisker pad and the dermal condensates of the whiskers (Supp. Fig. 5C).

At E11.5, mesenchymal complexity increases further, primarily reflecting the spatial organization of cells in the 3D space of the developing face (Fig. 3D). Within the LNP (*Pax7*), a proximal (*Irx5*) and a distal (*Casz1*) domains emerge (Supp. Fig. 6B, 6C1). In addition, *Gsc* and

*Gata2* expression, present in the LNP and MNP at E10.5, is restricted to the ventral nasal cavities lining the developing olfactory epithelium (Supp. Fig. 6C3). This ventral population (*Gsc*, *Gata2*), together with the dorsal domain of the MNP (defined by the expression of *Tfap2b* at E10.5), constitutes a molecularly heterogeneous region surrounding the nasal cavities, here referred to as the nasal cavity mesenchyme (Supp. Fig. 6C2–4).

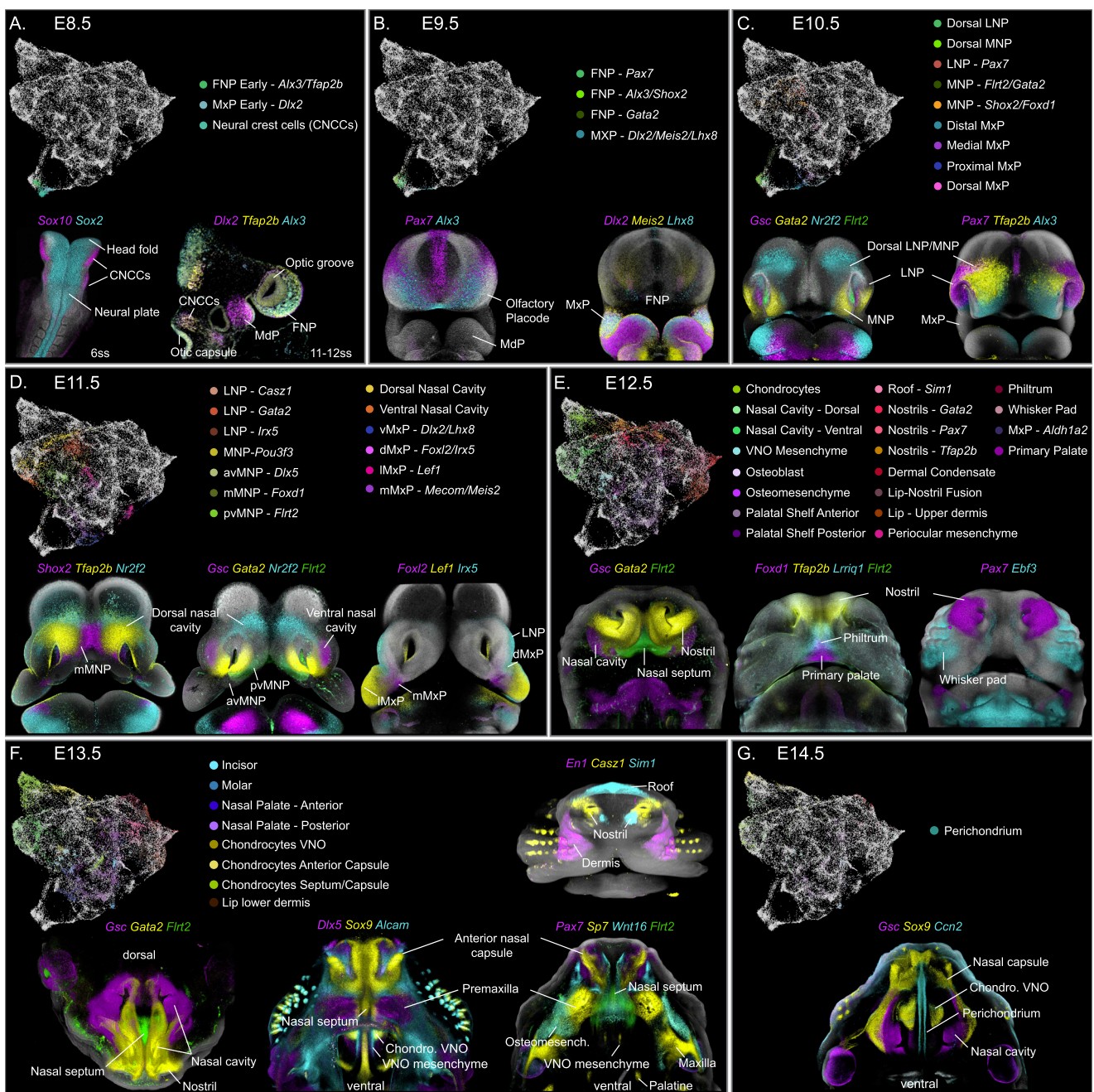

**Fig. 3 | Molecular heterogeneity of the developing face by embryonic stage. A-G** UMAPs visualization of the cell populations present at each developmental stage, from E8.5 to E14.5. The cluster names listed in each panel correspond to newly emerging cell populations at each stage. Multiplex in situ hybridization (HCR) for selected marker genes in each panel was used to confirm cluster identities and their localization in the 3D space. CNCCs cranial neural crest cells, FNP frontonasal prominence, LNP lateral nasal prominence, MdP mandibular prominence, MxP maxillary prominence, mMxP medial maxillary prominence, dMxP dorsal maxillary prominence, lMxP lateral maxillary, vMxP ventral maxillary prominence, MNP medial nasal prominence, avMNP anteroventral medial nasal prominence, pvMNP posteroventral medial nasal prominence, mMNP medial medial nasal prominence, VNO vomeronasal organ.

The MNP further splits into medial (mMNP), anteroventral (avMNP), and posteroventral (pvMNP) domains, each marked by the expression of distinct positional genes. The mMNP (*Foxd1, Cyp26b1*) is located towards the anterior midline, where the paired MNPs will fuse to form the philtrum (middle portion of the upper lip) and the primary palate (Supp. Fig. 6C6–9). The avMNP is marked by the expression of *Dlx5* and *Runx2* and indicates the position of the future premaxilla condensation (Supp. Fig. 6C5). Just behind the avMNP and surrounding the emerging vomeronasal organ is the pvMNP, which expresses *Flrt2* and *Sox9* and forecasts the location of the anteroventral nasal septum at E12.5 (Supp. Fig. 6C5–6). The expression of *Runx2* and *Sox9*

in the pvMNP and avMNP, respectively, marks the initiation of skeletogenic condensations in these areas and indicates the onset of cell fate commitment in the facial mesenchyme (Supp. Figs. 6B, 7C1).

At E11.5, the MxP undergoes further molecular segregation. The medial MxP (mMXP, *Mecom, Meis2*) additionally expresses *Shox2* and *Meox2* in the anterior and posterior areas, respectively. Both *Shox2* and M*eox2* are conventional markers of the palatal shelves[22,23], and their expression at this developmental stage forecasts the location and cellular source of these structures (Supp. Fig. 6C10). Interestingly, a localized expression of *Aldh1a2* marks a mesenchymal population surrounding the maxillary branch of the trigeminal nerve in proximity

to the presumptive palatal shelves (Supp. Fig. 6C13). The ventral MxP (vMxP) is marked by the expression of *Dlx2* proximally and *Lhx8* distally (Supp. Fig. 6C12). Part of the *Dlx2*-expressing vMxP overlaps with *Shox2* and *Dlx5* in the region where the temporomandibular joint will form[24] (Supp. Figs. 5C10, 6B). The ventral expression of *Lhx8* extends anteriorly up to the area of the lambda junction (Supp. Fig. 6C12). Additionally, part of the vMxP expresses *Tbx15*, a gene observed in the upper lip in later stages, indicating that this region contributes to the formation of this structure (Supp. Fig. 6C11). The lateral MxP (lMxP) emerges as a separate population underneath the facial epithelium expressing *Lef1*, forecasting its role in the formation of the dermis (Supp. Fig. 6C14). Finally, the dorsal region of the MxP (dMxP) expresses *Foxl2* and *Irx5* and remains located around the developing eye, marking the periocular mesenchyme (Supp. Fig. 6C14).

In conclusion, the comprehensive mapping revealed a temporal sequence of molecular events in the early stages of facial mesenchyme development, spanning CNCC migration, facial prominence formation and growth. During these stages, the remarkable molecular heterogeneity of the facial mesenchyme is driven by strong positional programs that correspond to the distinct physical positions of cell populations in the embryonic face and forecast the emergence of anatomical landmarks such as the upper lip, philtrum, and nostrils. These findings provide a high-resolution spatiotemporal molecular landscape of the developing face and suggest that spatial programs, rather than fate commitment, are essential in guiding facial morphogenesis, leading to the establishment of the first facial shape blueprint.

## Cell fate commitment and differentiation underlie the molecular heterogeneity of the late facial mesenchyme upon prominence fusion

At E12.5, the fusion of the facial prominences and the onset of cell differentiation introduce a new layer of molecular heterogeneity, complementing the positional information established earlier. While the clusters at later stages reflect the commitment to differentiation trajectories and newly formed structures, the derivatives of the LNP, MNP, and MxP largely retain the positional signatures (Fig. 3E–G), allowing for tracking the contribution of each facial prominence to the resulting facial structures. The molecularly distinct clusters of the late facial mesenchyme reflect the seven differentiation trajectories identified earlier (Fig. 2B, G). These emerging structures can be grouped as follows: chondrogenic lineage, nasal cavity mesenchyme, osteogenic lineage, palate, periocular mesenchyme and dermal mesenchyme, which includes the philtrum, the nostrils and the whisker pads.

Chondrogenic lineage: Chondrogenic condensations first appear in the frontonasal region at E11.5, specifically in the pvMNP and the dorsal nasal cavity mesenchyme (Supp. Fig. 7C1). By E12.5, these cells give rise to the nasal capsule (*Sox9, Pax3*) and the paraseptal cartilage surrounding the vomeronasal organ (*Flrt2, Sema3e*), each characterized by the expression of distinct positional markers (Supp. Fig. 7B, 7C2–3, 7C6). At E13.5, a third chondrogenic population, the anterior nasal capsule (*Wnt5a, Sp7, Msx1*), appears in the general cartilage cluster (Supp. Fig. 7C4, 7C7. A perichondrium cluster (*Ccn2*) emerges at E13.5 and is established by E14.5 (Fig. 3G; Supp. Fig. 7C7). In addition, an osteochondrogenic cluster (*Runx2, Sox9, Ccn2*), flanking the vomeronasal organ and the nasal septum, may represent the earliest sign of the vomer condensation (Fig. 3F; Supp. Figs. 7C5, 8C5).

Mesenchyme around the nasal cavities: Located between the cartilaginous nasal capsule and the nasal cavity epithelium, this mesenchyme arises from both the LNP and MNP. The mesenchyme around the nasal cavities molecularly segregates along the dorsoventral and anteroposterior axes and largely coincides with the position of the respiratory and olfactory epithelia. The nasal cavity mesenchyme, characterized by the expression of several positional genes, including *Gsc, Gata, Nr2f2*, and *Tfap2b*, will form the turbinates (Supp. Fig. 7B, 7C5–16).

Osteogenic lineage: The first osteogenic cells that emerge in the face are the premaxilla and maxilla condensations[25], which are derived from the avMNP and MxP, respectively (Fig. 3E, F; Supp. Fig. 8C1–4). Although the osteogenic mesenchyme and osteoblasts in both bone primordia express similar canonical bone marker genes (i.e., *Dlx5, Runx2, Sp7*), differential expression of positional genes in these clusters allows for discriminating the prominence of origin. For example, the osteoblasts of the premaxilla retain the expression of *Alx3*, an early positional marker of the MNP (Supp. Fig. 8C8), while the condensation of the maxilla expresses *Meis2*, an early marker of the MxP (Supp. Fig. 8C4, 8C19). Additionally, *Wnt16*, a gene participating in intramembranous ossification in the face[26], is observed in the osteogenic mesenchyme surrounding the cartilaginous nasal capsule at E12.5 and between the maxilla and premaxilla primordia at E13.5 (Supp. Fig. 8C1–2, 8C5). Interestingly, *Alcam*, a gene involved in osteoblast differentiation[27], is co-expressed with *Wnt16* in the osteomesenchyme (Supp. Fig. 8C2–4), as well as with *Sox9* in chondrocytes and perichondrium (Supp. Fig. 7B, 7C4).

Primary and secondary palate: The primary palate, the precursor of the premaxilla, forms from the fusion of paired MNPs at the facial midline (mMNP; *Foxd1, Cyp26b1*; Supp. Fig. 6C6, 6C8–9). At E12.5, the lateral and middle regions of the primary palate are defined by the expression of *Car2* and *Tbx22/Foxd1*, respectively (Supp. Fig. 8C9, 8C13, 8C16). By E13.5, a cluster representing the upper incisors (*Sfrp4, Car2*) emerges within the primary palate (Fig. 3F; Supp. Fig. 8C16). The secondary palate, the anlage of the palatal bones, arises from the palatal shelves derived from the mMxP (*Meis2, Mecom*; Supp. Fig. 6C12–13). At E12.5, the anterior and posterior domains of the vertically growing shelves are marked by *Shox2* and *Meox2*, respectively (Supp. Fig. 8C10–14). *Tbx22* further defines the dorsal area where the palatal shelves fuse with the primary palate (Supp. Fig. 8C10). The *Aldh1a2*-positive domain surrounding the maxillary branch of the trigeminal nerve is present in the osteogenic mesenchyme of the forming maxilla (Supp. Fig. 8C12). At E13.5, osteoblast differentiation (*Runx2, Sp7*) appears on the nasal side of the posterior palatal shelf, marking the location of the forming palatine bone (Supp. Fig. 8C5). Interestingly, the osteoblasts in the palatal shelves retain the expression of posterior palate positional markers (*Meox2, Sim2, Mecom, Pax9*), which distinguishes them from osteoblasts of other bone primordia (Supp. Fig. 8B). Additionally, a cluster representing the upper molar emerges within the posterior palatal shelf domain and is marked by the expression of *Tfap2b* and *Rgs5* (Fig. 3F; Supp. Fig. 8C20). *Shox2, Meox2*, and *Dlx5* are expressed in the location of the forming temporomandibular joint (TMJ, Supp. Fig. 8C10–11, 8C14), which derives from the posterior vMxP present at E11.5 (Supp. Fig. 6C12, 8C10).

Dermal Mesenchyme: The dermal mesenchyme (*Lef1*) encompasses four regions: the nostrils, the anterior roof of the snout, the whisker pads, and the sides of the upper lip (Fig. 3F; Supp. Fig. 9B). The molecular profile of the nostrils, which represent the dermal mesenchyme surrounding the anterior nasal vestibule, reflects the contributions from the LNP (*Pax7*) and MNP (*Pou3f3*), and it is further characterized by positional subdomains (Supp. Fig. 9C1–4, 9C14). The anterior roof of the snout (*Sim1*) is localized posterior to the nostrils above the nasal capsule (Supp. Figs. 8C6, 9C5) in the area where the nasal bone forms. It also expresses *Runx2, En1*, and *Osr1/Osr2* (Supp. Fig. 9B), thus likely corresponding to the earliest sign of the nasal bone condensation. The whisker pads (*Ebf3*) develop after LNP and MxP merge to form the cheeks. Most of the whisker pad retains *Meis2* in line with the known MxP origin[28] (Supp. Fig. 9C8). However, the upper anterior whisker condensate expresses *Pax7*, reflecting a partial contribution of LNP to sensory whisker formation (Supp. Fig. 9C8–9). The whisker pad cluster includes the dermal condensates of the whiskers (*Sox18, Foxd1, Sox21*) (Supp. Fig. 9C11–12, 9C15, 9C19). The upper lip is composed of two clusters: the philtrum (*Lrriq1, Foxd1*) originating from the mMNP (Supp. Fig. 9C6–7, 9C10, 9C12–13, 9C15, 9C18–19) and

the sides of the lip (*Lhx8, Tbx15, Tbx18*) derived from the vMxP and the lMxP (Supp. Fig. 9C12, 9C18–19). The sides of the lip comprise the upper and lower dermis (*Lef1/Tbx15* vs. *Dlk1/En1*)[21] (Supp. Fig. 9C16–17). Interestingly, although the nostrils, the lip and the whisker pads are all dermal structures, their positional profiles outweigh the cell fate signature and separate these cell populations into different clusters. Finally, *Foxl2*, a gene involved in maxillary and jugal bone development, as well as in eyelid formation[29], is consistently expressed in the periocular mesenchyme in the frontonasal and maxillary (dMxP at E11.5) regions (Supp. Fig. 10).

Altogether, detailed molecular characterization of facial development at later stages, upon the fusion of facial prominences (E12.5-E14.5; Supp. Fig. 11), revealed molecular heterogeneity driven by fate commitment and differentiation. However, the positional information from earlier stages is largely retained and contributes to the cluster identity, allowing for delineation of the cellular sources contributing to these structures. The generated scRNAseq data and gene expression mapping are available in a user-friendly, interactive online resource: https://www.evolbio.mpg.de/murillo-seton2025.

## Inference of cell communication along the facial developmental trajectory

Facial morphogenesis relies on bidirectional communication between ectodermal and mesenchymal cells. While signaling originating from the brain is known to initiate the head outgrowth and early patterning[30–32], and the facial ectoderm is acknowledged as a major source of morphogenetic signals, specifically driving facial shape acquisition in amniotes[33–35], the role of mesenchyme in this process is underexplored. We hypothesized that, in addition to the known ectoderm-derived signaling, individual mesenchymal populations may contribute to the facial morphogenesis by generating instructive cues. Here, we used CellChat[36] to predict intercellular signaling interactions based on the expression of ligand-receptor pairs in different cell populations. We applied this analysis to high-resolution clustering per developmental stage to uncover the dynamics of signaling utilization over time.

To explore the communication between mesenchymal and ectodermal populations, we included the ectodermal cluster (Fig. 1B, D) in the analysis. The ectoderm comprises the surface ectoderm (*Wnt6*) and its derivatives, such as the oral epithelium (*Pitx2*), periderm (*Grhl3*), and the nasal pit rim (*Fgf8*), as well as derivatives of the olfactory placodes (*Six1*), including neuronal (*i.e.* olfactory and vomeronasal epithelia, *Lhx2*) and non-neuronal (*i.e.* respiratory epithelium and supporting cells, *Foxa1, Maob*) lineages (Fig. 1D, F).

CellChat categorizes cell signaling based on the nature of the molecular interactions and classifies them into three main categories. 1) Extracellular matrix signaling (ECM) involves interactions between cell surface receptors and molecules of the extracellular matrix, such as collagen, fibronectin, and laminin. 2) Cell–cell contact signaling comprises interactions between neighboring cells through direct contact, mediated by cell adhesion molecules such as cadherins, integrins, nectins and junction adhesion molecules. 3) Secreted Signaling refers to communication mediated by molecules released into the extracellular space that bind receptors on target cells both in proximity and distance. Secreted signaling molecules include members of the BMP, WNT or FGF families, as well as Midkine and Pleiotrophin (MK/PTN) pathways. Due to the extent of MK/PTN signaling in early developmental stages, we also performed the analysis without MK/PTN to obtain better insights into the utilization of pathways from other categories (Fig. 4A vs. 4B). We found that the communication patterns are highly dynamic, employ signaling from all four categories, and their utilization trends correlate with the observed and known biology of the developing face (Fig. 4A, B; Supp. Figs. 12–18). For instance, our scRNAseq analysis showed high proliferation rates in the facial mesenchyme (Supp. Fig 1H), particularly between E8.5 and E10.5.

This aligns well with the dominating activity of MK/PTN growth factors, representing more than 50% of the predicted interactions, both in the ectoderm and mesenchyme (Fig. 4A, blue bars). Cell-cell contact (Orange bars) and ECM-receptor (Green bars) signaling generally increase with development progression (Fig. 4B). Between E8.5 and E11.5, the mesenchyme is less involved in cell-cell and ECM contact than ectoderm (Fig. 4B darker vs paler color shades), possibly allowing the cells to engage in viscous-fluid-like behavior and crowd movement resulting from extensive proliferation, in line with previously reported findings[13,37,38]. At E12.5, the stage displaying clear molecular signs of cell fate commitment (Fig. 2C–G), particularly in the chondrogenic, osteogenic and dermal lineages, exhibits an increase of ECM-Receptor signaling, primarily composed of the Collagen and Integrin family is observed (Fig. 4A, B; Supp. Figs. 16–18).

CellChat's inference of cell-cell interactions allows for the identification of major signaling sources (i.e., senders; cells releasing signaling molecules, such as a secreted ligand), and their targets (i.e., receivers; cells that express corresponding receptors and respond to a released signaling molecule) (Supp. Data 1). The response of the receiver cells to the released molecule is referred to as "incoming signaling", while the release of molecules by a sender is defined as "outgoing signaling". As such, senders are the drivers of outgoing signaling, while receivers are responsible for incoming signaling.

In particular, secreted signaling is known to control cell behavior and patterning across developing complex structures[39]. We found that ectodermal populations act as primary senders of this signaling type in the early developmental stages (Fig. 4B, outgoing signaling, purple pale bars; Fig. 4C–E11.5; Supp. Figs. 12–18). Specifically, the midfacial ectoderm located in between the olfactory placodes at E9.5 (Figs. 4D1, 5A; Supp. Fig. 13B), as well as the nasal pit rim at E10.5 and E11.5, representing the transition zone between the olfactory placode (*Six1*) and surface ectoderm (*Wnt6*) (Fig. 4D2, Fig. 5B, C, F, G), are prominent sources of secreted signaling (Supp. Figs. 14B, 15B; Supp. Data 2, 3), suggesting a central role for these populations in orchestrating facial morphogenesis prior to prominence fusion. Outgoing secreted signaling from the nasal pit rim (Fig. 4C–E11.5, arrow) primarily comprises BMP, FGF and non-canonical WNT signaling components (Fig. 5; Supp. Figs. 19C, E, G, 20; Supp. Data 2, 3). This prediction is supported by previous studies demonstrating the essential role of the nasal pit rim as a morphogen source in face formation across species[40–42]. On the contrary, at these early stages, the mesenchymal populations generally act as target populations (i.e., receivers) of secreted signals (Fig. 4B, incoming signaling, dark purple bars; Supp. Fig. 19). Interestingly, upon facial prominence fusion at E12.5, the mesenchymal populations located in the front of the forming face (i.e., the nostrils, philtrum, and lip-nostril fusion domains, Fig. 4C–E12.5, oval) become a main source of secreted signaling (Fig. 4B, outgoing signaling, dark purple bars), primarily employing non-canonical WNT and BMP (Fig. 5D, H; Supp. Figs. 19D, 19H, 20; Supp. Data 2, 3). This switch in the production of secreted morphogens suggests that mesenchymal populations may act as late signaling centers for face morphogenesis.

In summary, CellChat analysis uncovered the communication dynamics between ectodermal and mesenchymal populations across stages, predicting communication tools employed by distinct cell populations and identifying sources (e.g., signaling centers) with their potential target populations. The inference aligns with the observed cellular and molecular dynamics (Fig. 2; Supp. Fig 1H, L) and previously described morphogenetic events, highlighting how cell populations coordinate their behavior during development to create a functional facial morphology. While the results align with the known role of ectoderm in morphogenetic signaling, we revealed an unappreciated role of facial mesenchymal populations in generating instructive cues, particularly from E12.5 onwards. In addition, this analysis predicts the involvement of pathways and signaling centers that have not been

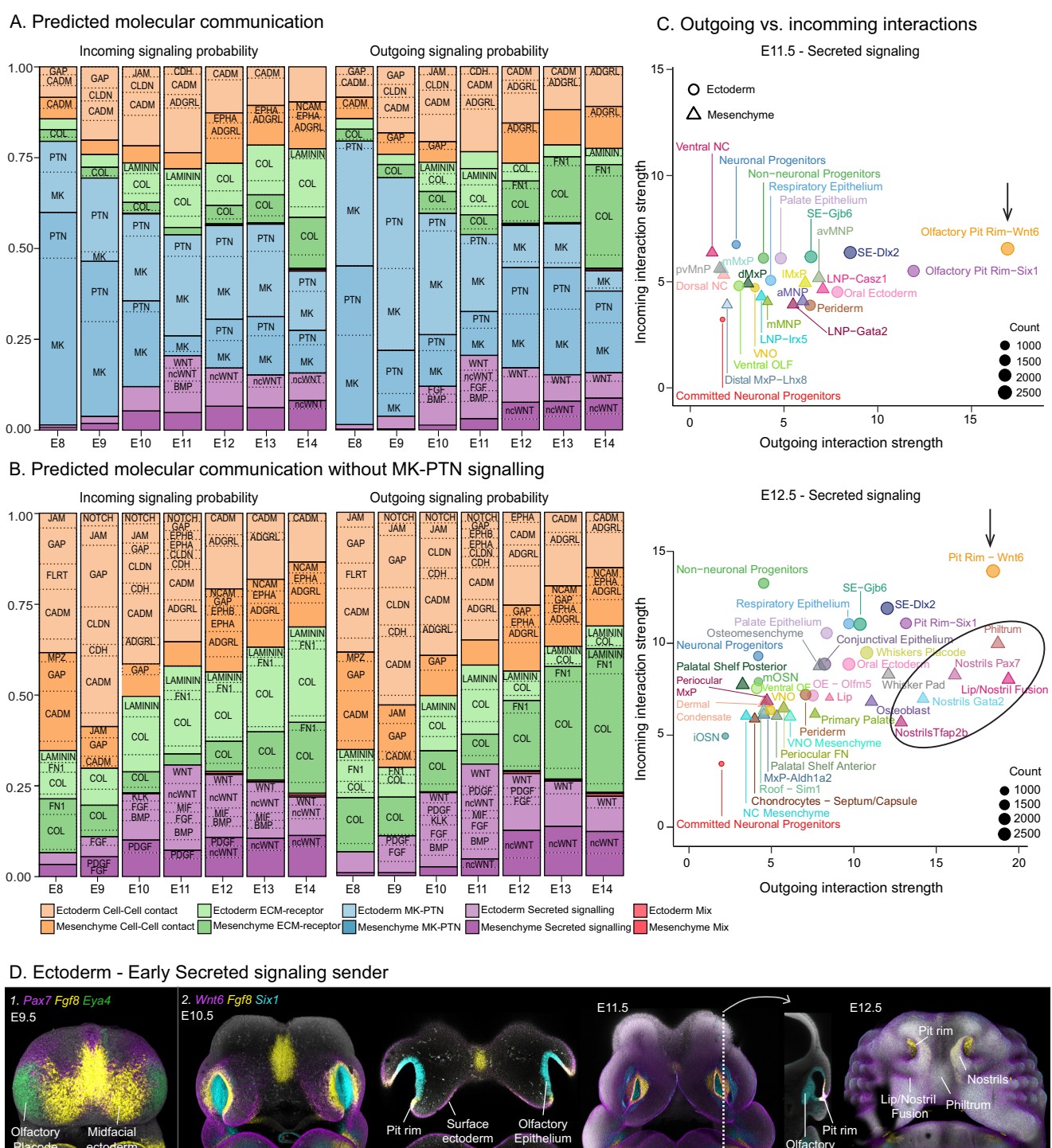

**A. Predicted molecular communication**

**B. Predicted molecular communication without MK-PTN signalling**

**C. Outgoing vs. incoming interactions**

**D. Ectoderm - Early Secreted signaling sender**

extensively studied in facial morphogenesis, such as members of the adhesion G-protein-coupled receptor family (e.g., ADGR, Fig. 4), a major contributor to cell-cell adhesion interactions, and serves as a basis for future targeted investigations.

**Cellular sources of natural human facial shape variability**

Human facial shape is a highly complex, polygenic, heritable trait, and its formation is orchestrated by intricate molecular networks[43,44]. Genome-wide association studies (GWAS) have provided a wealth of information on genetic loci associated with natural, non-syndromic variability in facial morphology, as well as facial congenital disorders.

Despite the volume of available data, interpreting these findings and linking them to underlying mechanisms has been challenging. Additionally, a long-standing question is whether facial shape variation is mainly generated by alterations of ectoderm-derived morphogen gradients or the local ability of mesenchymal cells to respond to these instructive signals[45]. Here, we aimed to fill these gaps by integrating the reconstructed developmental history of the murine face with the extensive repertoire of existing human GWAS studies to reveal the developmental origins of human facial shape diversity.

To investigate which cell populations contribute the most to natural human facial shape variation, we performed gene set

**Fig. 4 | Predicted cellular communication between ectodermal and mesenchymal populations. A** Stacked bar plot showing the relative signaling probability of the four signaling categories analyzed per stage. Signaling or communication probability refers to the predicted likelihood of two cell populations interacting through a given signaling pathway, based on the expression levels and known interactions among receptors, their ligands and cofactors. Differences in predicted contribution to incoming (left) and outgoing (right) signaling between ectoderm and mesenchyme are shown. Colors represent different signaling categories. Light and dark shades of each color represent ectoderm and mesenchyme, respectively. The dotted lines show the proportion of individual signaling pathways comprising each signaling category. Names of signaling pathways contributing >2% to the total signaling probability are shown. Signaling pathways contributing to <2% of the total signaling probability are grouped, and no name is shown. **B** Stacked bar plots showing the relative signaling probability similar to (**A**) but excluding MK and PTN pathways. **C** Scatter plots for predicted incoming and outgoing secreted signaling probability per cluster at E11.5 (above) and E12.5 (below). Clusters with higher incoming or outgoing interaction strength values are considered stronger receiver or sender populations, respectively. Interaction strength refers to the magnitude and probability of the predicted interaction, where the higher the strength, the higher the probability of signaling taking place between cells. Circles and triangles represent ectodermal and mesenchymal populations, respectively. Shape size is proportional to the number of predicted signaling links (both outgoing and incoming) associated with each cluster; in other words, shape size reflects the number of interactions predicted, regardless of their magnitude. Black arrows point to the nasal pit rim ectodermal population, the primary sender of secreted signaling at early stages. The black circle highlights the mesenchymal populations that take over as the main secreted signaling senders at E12.5. Corresponding *p*-values for all displayed interactions are presented in Supp. Data 1. Only significant interactions are displayed in the plot. Interaction probabilities are presented in Supp. Data 2 and 3. **D** In situ hybridization (HCR) showing secreted ligand expression in the 1) midfacial ectoderm at E9.5 and the 2) nasal pit rim at E10.5, E11.5 and E12.5. Images are not in scale. ADGRL adhesion G-protein-coupled receptor L1, CADM cell adhesion molecule, BMP bone morphogenetic protein, CDH cadherin, CLDN claudin, COL collagen, EPHA/B Ephrin A/B, FGF fibroblast growth factor, FLRT fibronectin leucine-rich repeat transmembrane, FN1 fibronectin 1, GAP gap junctions, JAM junctional adhesion molecules, KLK kallikrein-related peptidase, MIF macrophage migration inhibitory factor, MPZ myelin protein zero, NCAM neural cell adhesion molecule, ncWNT non-canonical WNT, PDGF platelet-derived growth factor, WNT wingless-related integration site.

enrichment analyses using AUCell. This analysis examines whether a specific set of genes is overrepresented among genes expressed by a certain cell population. First, we utilized the extensive catalog of genetic variants from the OpenTargets platform database to compile a list of genes (gene set) associated with genetic variants linked to human facial shape variation under the Experimental Factor Ontology (EFO) term Facial Morphology Measurements (Supp. Data 4). The retrieved gene set comprised 140 genes.

We then integrated this gene set with single-cell data from mouse to systematically examine the enrichment of genes linked to natural facial shape variation across cellular populations and embryonic stages (Fig. 6A). Interestingly, this analysis revealed a strong enrichment in the mesenchyme compared to other cell types (Fig. 6B; Supp. Fig. 21). We did not detect any notable enrichment in brain and ectoderm, indicating that the cellular and molecular sources of natural human facial shape variation are primarily harbored in the mesenchymal populations.

When examining the mesenchymal subset specifically (Fig. 6C), the mean relative AUCell score per stage indicated that later developmental stages contributed more prominently to human facial shape variability than earlier stages (Fig. 6D). Within the mesenchyme, the facial morphology gene set was particularly enriched in dermal clusters such as the nostrils, the snout roof, philtrum and lip (Supp. Fig. 21). Notably, these frontal facial areas are known to have high shape heritability[44].

Importantly, no enrichment was detected in randomly selected gene sets (Supp. Fig. 22B). Furthermore, gene sets linked to EFO terms for other traits showed enrichment in expected cell types, serving as controls (Supp. Fig. 22C–F; Supp. Data 5). For example, EFO terms Generalized Epilepsy, Immune system disease, and Keratosis show clear enrichment in the clusters of brain, immune cells and ectodermal cells, respectively, supporting the validity of these findings.

To identify the cellular sources of feature-specific geometries, we used the LMD, a score indicating how restricted the expression of a given gene is to a specific area of UMAP (i.e., whether a gene is specific to a cell population). Out of the 140 genes from OpenTargets that are associated with normal human facial variation, 41 genes showed a particularly spatially restricted expression pattern in the mouse developing face (Supp. Data 4, values marked in green). Among these 41 spatially restricted genes, 22 harbor 65 variants associated with human facial traits that map to corresponding anatomical regions in mice (Supp. Data 4, compare green and yellow values). For instance, in the mouse, *Casz1*, *Gata2* and *Lrriq1* are molecular hallmarks of the mesenchyme surrounding the nostrils dorsally and ventrally, and

expressed in the philtrum, respectively (Fig. 7A–C). In humans, genetic variants in *Casz1* and *Gata2* are linked to nostril width, while *Lrriq1* variants are associated with philtrum width[46–49]. Notably, *Lrriq1* has also been associated with beak shape variation in birds[50,51], suggesting that face-shaping molecular programs may be conserved across amniotes. These findings highlight how spatially restricted populations not only play a role in face development but also carry genetic signatures linked to naturally occurring intraspecific phenotypic diversity.

Additionally, *Pax3*, a gene consistently associated with human facial shape variation, particularly with the position of the nasion, the deepest point of the nasal bridge (Fig. 7D), harbors variants implicated in both normal and abnormal shape variation in humans[48,52–54]. Mutations in *Pax3* are associated with Waardenburg syndrome type I, a rare neurocristopathy characterized, among others, by the widening of the nasal bridge area (Fig. 7H)[55–57]. Therefore, the effect of *Pax3* on facial shape variation has been previously attributed to the CNCCs[48,58]. Although we detected *Pax3* in the CNCCs cluster, a strong expression can be observed at later developmental stages, particularly in the LNP (E11.5), nostrils (E12.5), nasal cavity mesenchyme and around the nasal capsule walls (E13.5) (Supp. Fig. 23), suggesting that the influence of *Pax3* on normal facial shape variation primarily results from the mesenchyme rather than the CNCCs biology. Although *Pax3* localization is less spatially restricted than that of *Lrriq1* or *Casz1*, its expression in the mouse snout aligns well with the observed nose shape variation in humans.

To investigate which cell populations are most involved in abnormal craniofacial development (craniofacial disorders), we compiled a gene set comprising 89 genes known to cause or contribute to craniofacial abnormalities in humans (DISGENET, see methods for terms) (Supp. Data 6). This gene set also showed an enrichment in the mesenchyme compared to other cell populations (Fig. 6E), likely reflecting the fact that disruptions of key developmental genes are often embryonically lethal and thus underrepresented in human craniofacial abnormalities databases. Therefore, we compiled a list of 48 genes associated with Abnormal Facial Prominence Development in the mouse (Mouse Genomic Informatics, MGI database) (Supp. Data 7). AUCell analysis revealed that the mouse gene set was enriched in the mesenchyme, ectoderm, and brain clusters (Supp. Fig. 24), consistent with their established role in facial morphogenesis. Additionally, we explored whether the same cell populations are involved in both normal and abnormal facial shape variation. Individual mesenchymal populations are differentially enriched for normal and abnormal gene sets. For example, chondrocyte populations are highly enriched in

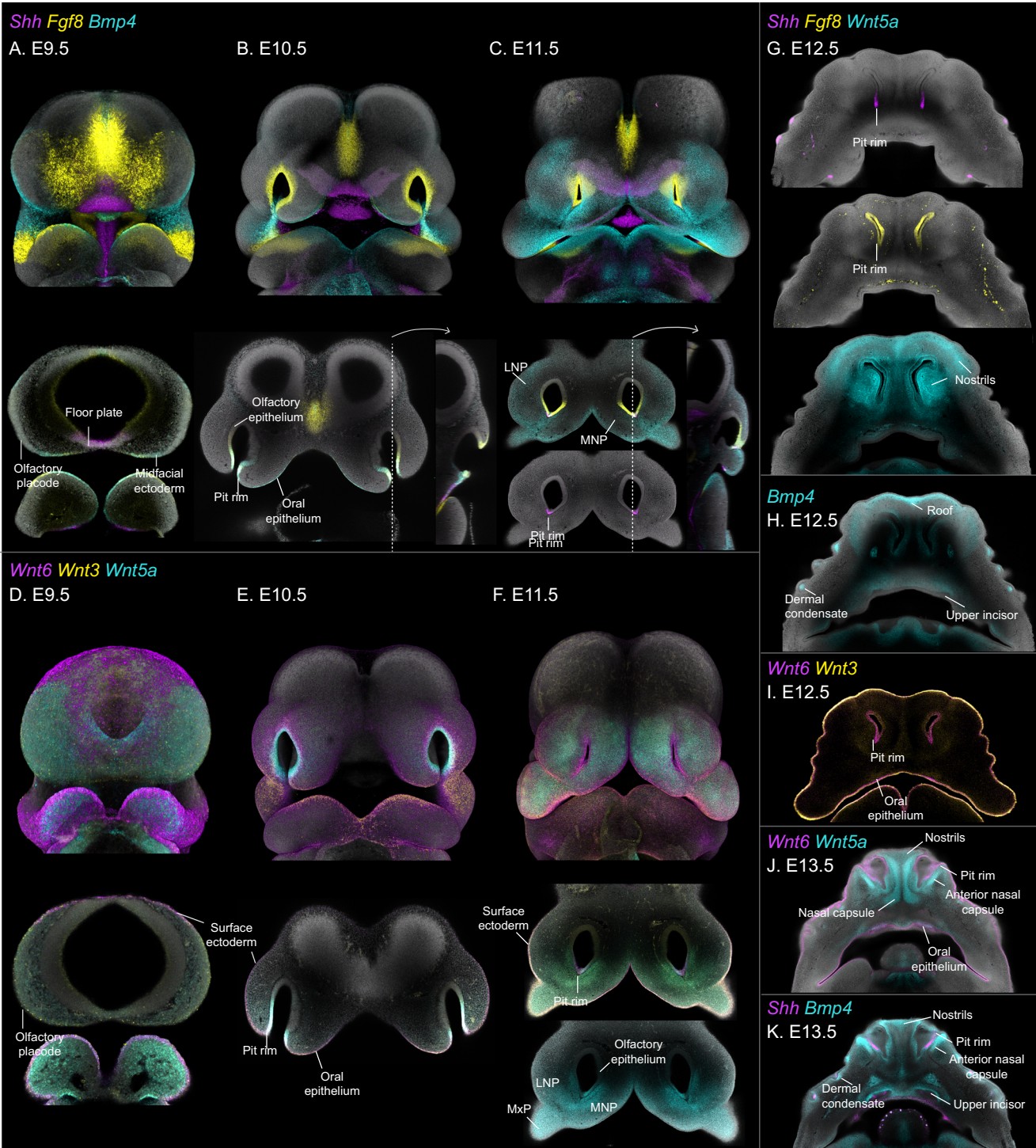

**Fig. 5 | Morphogen expression shifts from ectodermal to mesenchymal populations between E11.5 and E12.5. A–C** Sonic hedgehog (*Shh*), Fibroblast growth factor 8 (*Fgf8*) and Bone morphogenic protein 4 (*Bmp4*), as well as **D–F** canonical Wingless proteins 6 (*Wnt6*) and *Wnt3*, and non-canonical *Wnt5a* expression in whole-mount and frontal sections of embryonic heads at E9.5 (**A**, **D**), E10.5 (**B**, **E**), E11.5 (**C**, **F**), E12.5 (**G–I**) and E13.5 (**J**, **K**). Through the analyzed stages, *Shh* and *Fgf8* expression was restricted to epithelial tissues, such as the midfacial ectoderm, brain floor plate, nasal pit rim and oral epithelium. Although *Bmp4* expression is also confined to the epithelium at early stages, at E11.5, the expression appeared

also in the facial mesenchyme. At E12.5 and E13.5, *Bmp4* expression is observed in the mesenchyme around the nostrils, the upper incisors, and in the dermal condensates (**H**, **K**). Similarly, while *Wnt6* and *Wnt3* were consistently expressed in the surface ectoderm, oral epithelium and pit rim from E9.5 to E12.5, *Wnt5a* was observed first in the pit rim and olfactory epithelium, and then in the facial mesenchyme from E11.5 onwards (**G**, **J**). Broken lines in **A**) mark the level of the section shown on the right, indicated by an arrow. FNP frontonasal prominence, LNP lateral nasal prominence, MNP medial nasal prominence, MxP maxillary prominence, MnP mandibular prominence.

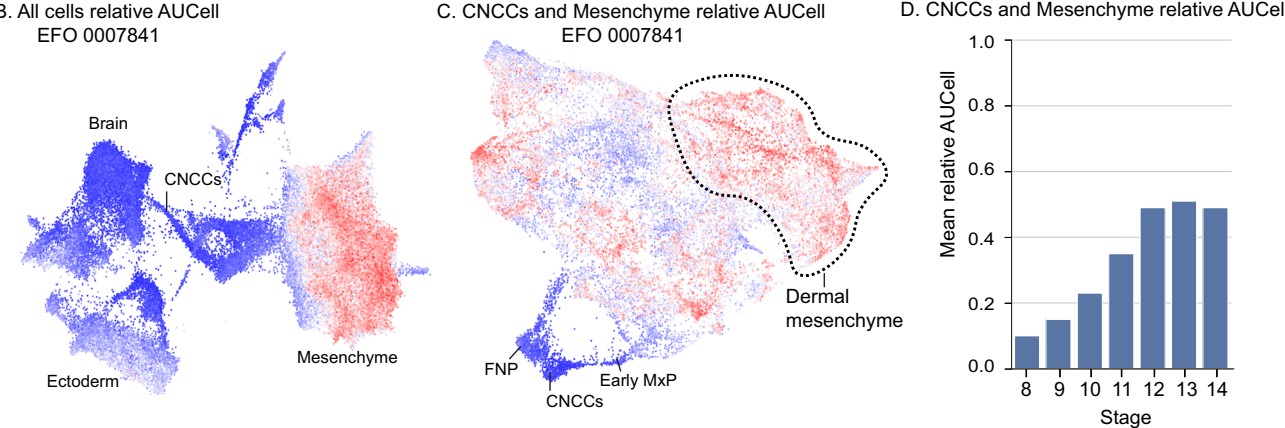

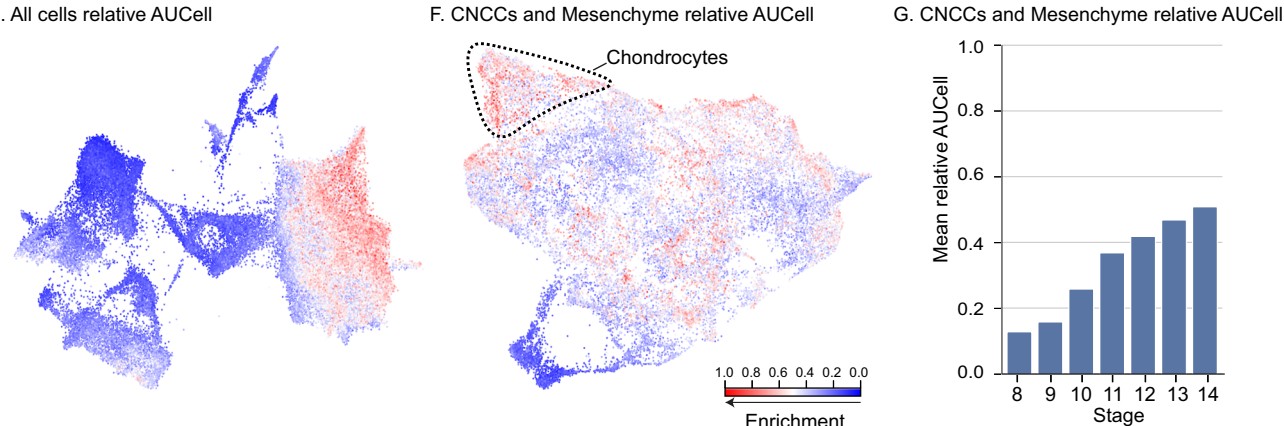

**Fig. 6 | Identification of cellular sources of human natural facial shape variability. A** Schematic representation of the methodological approach. The Open Targets Platform database was used to compile genes associated with human facial shape variation from GWAS studies. AUCell was applied to define whether the obtained gene sets were enriched in specific clusters of the single-cell RNA-seq data from Fig. 1B. Relative AUCell scores (normalized between 0 and 1) were used to identify bimodal distributions and highlight enriched cells. **B** Relative AUCell scores for genes associated with EFO_0007841 (facial morphology measurement, Supp. Data 4) visualized on the integrated UMAP for all cells analyzed (Fig. 1B). Note that mesenchymal cells exhibit high enrichment scores. **C** Relative AUCell scores for genes associated with EFO_0007841 (facial morphology measurement), visualized

on the UMAP for neural crest cells and facial mesenchyme (Fig. 1C). **D** Mean relative AUCell score for genes associated with EFO_0007841 (facial morphology measurement) per stage in the cranial neural crest cells (CNCCs) and facial mesenchyme populations, highlighting an increased enrichment of EFO_0007841-associated genes at later stages (E11.5-E12.5). **E–G** Relative AUCell scores for genes associated with abnormal craniofacial development in humans, retrieved from the DISGENET database (Supp. Data 6), visualized on the UMAP for **E** all cells and **F** cranial neural crest cells and facial mesenchyme only. **G** Mean relative AUCell score for genes associated with abnormal facial prominence development per stage in CNCCs and facial mesenchyme populations. The human face icon in A was created in BioRender. https://BioRender.com/ms2nr5s.

genes associated with human craniofacial abnormalities, but much less in genes linked to natural facial shape variation (Supp. Figs. 21, 25), highlighting the role of different cell populations in fundamental versus fine-tuning developmental processes.

To further support the role of positional programs in the formation of human facial features and the utility of mouse-to-human data extrapolation, we manually searched for the role of murine early positional genes identified in our dataset in establishing both

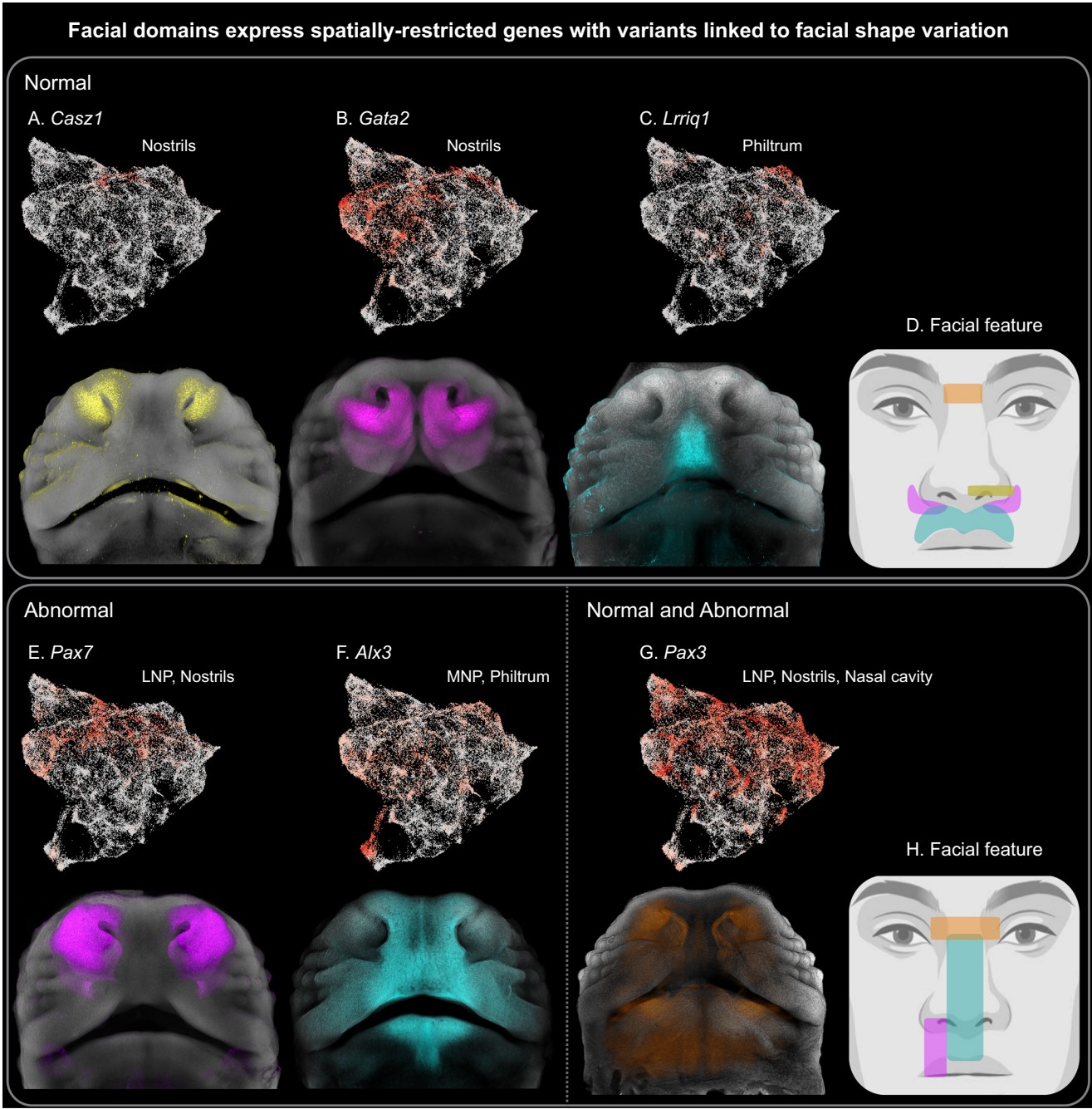

**Fig. 7 | Facial domains express spatially restricted genes that carry variants associated with normal and abnormal facial shape variation. A–C** Selected late positional genes marking specific facial structures are associated with natural face variation in humans, identified in GWAS studies. **A** *Casz1*, rs143353512, **B** *Gata2*, rs7373685 (Facial Segment 18 - Upper lip/philtrum) and rs2977562 (Facial Module 16 - Upper lip/philtrum). **C** *Lrriq1*, rs11116771 (Quadrant I - Philtrum, Segment 37). **D** Scheme indicating the corresponding facial features associated with *Casz1* (yellow), *Gata2* (magenta), *Lrriq1* (blue) and *Pax3* (orange) variants. **E, F** Early positional genes underlying congenital conditions characterized by abnormal facial shape features in humans (Supp. Data 8). E) *Pax7* is an early positional marker of the LNP

associated with cleft lip in humans. F *Alx3* is an early positional marker of the MNP associated with frontonasal/midfacial dysplasia in humans. **G** *Pax3* is expressed in the NCCs and the facial mesenchyme from E11.5. *Pax3* variants are associated with normal and abnormal nose shape variation in humans. **H** Scheme indicating the corresponding facial regions affected by *Pax7* (magenta), *Alx3* (blue), *Casz1* (yellow) and *Pax3* (orange) in humans. UMAPs and in situ hybridization (HCR) in A-C and E-G display gene expression domains of corresponding genes in E12.5 embryonic faces. The human face icon in D and H was created in BioRender. https://BioRender.com/k6pu8s9.

murine and human facial morphology (Supp. Data 8). We found that disruption of early positional genes leads to abnormal facial phenotypes or embryonic lethality. For instance, *Alx3* and *Pax7*, early positional marker genes of the MNP and LNP in mouse, respectively, are linked to the abnormal facial shape features observed in homologous areas in the human face, such as

frontonasal dysplasia and non-syndromic cleft lip, respectively (Fig. 7E–G; Supp. Data 8)[59,60].

Overall, these analyses highlight the central role of the facial mesenchyme in generating natural facial shape variation in humans. By integrating mouse single-cell RNA sequencing with an extensive repertoire of genetic variants from human GWAS studies, we

established a framework for linking developmental programs and cell populations to genetic variants associated with phenotypic diversity. This work also provides valuable insights into the effector cell populations contributing to congenital craniofacial disorders, providing a foundation for investigating developmental disruptions that cause structural abnormalities. Moreover, this approach represents an efficient alternative to time- and resource-intensive in vivo functional experiments that allow for testing of a limited number of genes in an experiment.

## Discussion

Face formation and shape acquisition result from an intricate interplay of genetic, epigenetic, and molecular mechanisms and require coordinated interactions between multiple tissues, particularly the developing brain, ectoderm, and facial mesenchyme[61]. However, the mechanisms linking these levels, specifically how genetic information translates into distinct facial shapes through the establishment of cell-specific transcriptomes and cell behavior, are not yet fully understood. While the previous research on facial development has mainly focused on the ectoderm and brain[30,31,61], the role of the mesenchyme remains comparatively underexplored. The mesenchyme builds the skeletal structures of the face and ultimately dictates the facial shape; hence, our focus is more directed towards elucidating the molecular programs operating within the facial mesenchyme. In this study, we aimed to reconstruct the developmental history of the face and explore whether we could leverage this information to gain better insights into the origins of natural facial variability and its cellular and molecular underpinnings.

The human face is a multipartite structure composed of features with distinct heritability values[44,62]. The differential genetic control exerted over the local geometry of individual facial features suggests the existence of mechanisms allowing for a localized manifestation of genetic information, subsequently leading to the individual shapes of facial features. Additionally, the independent heritability of facial features, for example, inheriting the shape of the nose from one parent and the lips from another[62], supports the concept of area-specific manifestation of genetic background. Therefore, we specifically sought evidence for the existence of such position-specific developmental programs that would permit localized phenotypic expression of genetic information. Here, we propose that the positional programs of the facial mesenchyme may underpin the independent heritability of discrete facial features.

We revealed that the remarkable heterogeneity of the facial mesenchyme is generated in two distinct phases of molecular segregation. From E9.5 to E11.5, the molecular heterogeneity of facial mesenchyme is primarily driven by positional information. The molecular profiles of individual mesenchymal populations reflect their positions within facial prominences and exhibit even more spatial and molecular separation within these regions. These spatial subdomains and their unique molecular signatures forecast the emergence and positioning of distinct facial structures, suggesting that these early spatial entities serve as cellular precursors for specific facial structures. Furthermore, even after cells commit to or differentiate into specific cell types (e.g., osteoblasts, chondrocytes, or dermal fibroblasts), they retain the expression of early positional markers. This allowed us to determine the origin of the cells building the same tissue but in different locations of the developing head. For instance, osteoblasts that form the maxilla, premaxilla, and palatal bone share the expression of canonical markers (*Runx2, Sp7*); however, the cells also express either *Meis2, Alx3* or *Meox2/Sim2*, allowing us to assign them to these distinct bone primordia, respectively. Similarly, chondrocytes (*Col2a1, Sox9*) are further split into *Wnt5a/Sp7*, *Flrt2/Dlk1* and *Pax3*, indicating their contribution to the anterior nasal capsule, vomeronasal and the rest of the nasal capsule cartilage elements, respectively. Additionally, the dermal condensates

(*Lef1*) derived from the LNP and MxP can be distinguished by the expression of *Pax7* and *Meis2*, respectively.

The second phase of molecular heterogeneity begins at E12.5 and is characterized by differentiation-driven separation as mesenchymal cells commit to specialized lineages, such as chondrocytes and osteoblasts. This phase additionally exhibits changes in proliferation rates and intercellular signaling. In particular, there is an evident shift in the predicted secreted signaling, in particular, of BMP and non-canonical WNT pathways, between ectodermal and mesenchymal populations (Fig. 4A–C). During earlier stages, ectodermal populations, such as the pit rim and the oral epithelium, act as prominent signaling centers, releasing key instructive molecules from the *Fgf*, *Bmp* and *Hh* gene families[34,35,42] (Fig. 4C; Supp. Fig. 19). However, at later stages, mesenchymal populations, particularly in the anterior part of the face, such as the nostrils and the philtrum, become the primary source of BMP and non-canonical WNT signaling (Fig. 4C; Supp. Fig. 19). Interestingly, disruptions in BMP signaling between E12.5 and E14.5 have been previously linked to changes in facial length in mouse, while the overall face morphology was preserved[63,64]. Similarly, ablation of *Wnt5a* in the NCCs mainly affects the elongation of the face, while the rest of the head remains largely unaltered[65,66]. According to our predictions and gene expression analysis, the mesenchyme becomes the main source of non-canonical WNT/*Wnt5a* signaling from E11.5 onwards (Figs. 4B, 5E–H). While the effect of the mesenchyme-derived morphogenetic signals on facial morphogenesis in later stages (from E12.5 onwards) has to be further explored functionally, the available information and our results suggest that the frontal mesenchymal populations may act as late developmental organizers, contributing to fine-tuning the morphology of frontal facial features[63,64]. Differential expression of such conserved genes has been associated with more prominent changes in face morphology, as observed in inter-specific facial shape variability or pathological conditions[67–72]. Similarly, pharmacological or genetic disruptions of ectoderm- and brain-derived morphogens result in larger changes in geometrical proportions and features[13,31–33]. However, an important question remains: when do relatively minor variations in individual facial features emerge, such as in humans, and which cells are responsible for generating this intraspecific variability? Therefore, to identify the developmental stages and cell populations contributing to natural facial variability in humans, we integrated the reconstructed single-cell developmental trajectories found in mice with existing information from GWAS studies on craniofacial morphology in humans. Interestingly, genetic variants associated with normal facial variation are strongly enriched in genes highly expressed by mesenchymal populations, particularly from E11.5 onwards, suggesting a later developmental origin of facial shape diversity. Thus, the CNCCs (E8.5) and early facial mesenchyme (E9.5–10.5) are likely less prominent drivers of normal human facial variability, contrary to previous suggestions[48,58]. We found a striking correlation between the genetic variant, the reported trait and the location where the gene is highly or specifically expressed, suggesting that the positional programs acting throughout the face development allow for the localized manifestation of genetic information. For instance, *Casz1* is expressed by the mesenchymal population surrounding the developing nostrils, and the genetic variant of this gene is associated with nostril breadth in humans[46]. Similarly, the genetic variant in *Lrriq1*, a gene expressed in the mesenchyme forming the philtrum, is associated with the morphology of this region[49]. Additionally, we searched for genes reportedly linked to abnormal facial prominence development in the mouse (Supp. Data 6). These genes were enriched in three major populations: the brain, the mesenchyme, and the ectoderm (Supp. Fig. 24A, B). This enrichment pattern is expected, considering the established roles of brain- and ectoderm-derived signals and mesenchyme patterning in craniofacial development[30,31,73]. Moreover, we found that perturbations in murine positional genes expressed by early mesenchymal

populations (e.g., *Alx3, Pax7, Tfap2b*) are closely associated with abnormal facial features such as cleft lip/palate or frontonasal dysplasia in humans (Supp. Data 8)[59,60]. Therefore, we conclude that while the mesenchyme in association with the brain and the ectoderm gives rise to large-scale geometric changes, such as observed in interspecific variation and pathological conditions, mesenchymal populations at later stages, including cells undergoing cell commitment, are likely to influence the local fine-tuning of individual facial features.

In conclusion, by investigating the developmental history of the face on a single-cell level and high-clustering resolution, we identify the cellular and molecular underpinnings of natural facial shape variability. The findings also contribute to our understanding of abnormal facial variation, suggesting that disruptions in spatially defined mesenchymal populations can lead to defects in specific facial features, such as frontonasal dysplasia. This knowledge will guide future investigations into the etiology of craniofacial syndromes, allow us to identify causative cell populations and trace the molecular mechanisms underlying structural abnormalities. The interactive database of transcriptomic profiles with extensive HCR imaging data produced in this study is made publicly available at https://ome. evolbio.mpg.de/. Finally, this study lays the foundation for comparative and evolutionary research. Investigating positional heterogeneity of facial mesenchyme across species and identifying homologous populations and their existence along embryonic space and time will shed light on the role of mesenchyme in both microevolutionary and macroevolutionary changes in facial morphology, further exploring the basis of facial morphological evolution.

## Methods

### Animal information

All mouse work was performed in the animal facility of the Max Planck Institute for Evolutionary Biology in accordance with Directive 2010/63/EU, the German Animal Welfare Act (Tierschutzgesetz § 11) and the guidelines of the Federation of European Laboratory Animal Science Associations (FELASA), under a 12:12 light:dark cycle (room temperature 20–24 °C), humidity 45%–65%. Mouse embryo collection during the second week of gestation (E8.5–E13.5) was performed in accordance with the German Animal Welfare Act (§ 4 (3) TierSchG). The collection of embryos at the 3rd week of gestation (E14.5) was approved by the Ministerium für Landwirtschaft, ländliche Räume, Europa und Verbraucherschutz des Landes Schleswig-Holstein, under permit 53-6/20.

### Embryo collection and fixation

Wild-type C57BL/6 N mouse embryos (both males and females), from three independent litters per stage, 3–10 embryos per stage, were used in this study. Breeding pairs were mated overnight, and the presence of a vaginal plug was assessed the following morning. The noon of the day of the detected plug was considered embryonic day E0.5. Females were monitored for weight gain. Pregnant mice were sacrificed by cervical dislocation, and embryos were dissected under a stereomicroscope in sterile ice-cold 1× Dulbecco's Phosphate Buffered Saline (PBS) (Sigma D5652). Embryos were fixed in freshly prepared 4% paraformaldehyde (PFA) in PBS at 4 °C for 1–16 h on slow rotation, depending on the developmental stage and experimental application. For whole-mount immunofluorescence and HCR in situ hybridization, samples were dehydrated in increasing methanol steps 10 min each (25%, 50%, 75% and 100%) in 0.1% PBS-Tween (PBST) (Tween 20; Sigma P9416) at 4 °C on a slow rotation and stored in methanol at −20 °C till further use. Mouse embryos for single-cell transcriptomics were processed according to the protocol described below.

### Sample preparation for single-cell transcriptomics

After dissection from the uterus, the developmental stages were additionally confirmed by the assessment of the known morphological landmarks. The upper face of E10.5, E11.5, E12.5, E13.5 and E14.5 embryos (lateral nasal, medial nasal and maxillary prominences and the structures derived from them), excluding the eyes, were microdissected and placed in ice-cold PBS. The skin of E13.5 and E14.5 embryos was partially removed to enrich the presence of mesenchyme. Dissected upper faces from multiple embryos of the same stage were pooled to secure an optimal number of cells in the sample. Collected tissue was centrifuged at $500 \times g$ for 5 min at 4 °C, and PBS excess was removed and replaced by 1 mL of pre-warmed trypsin (0.05% Trypsin/0.02% EDTA; Pan Biotech P10-0235SP), incubated at 37 °C for 15 min with gentle shaking and gently pipetted for 30 seconds (s) every 5 min. Following enzymatic treatment, the tissues were further dissociated by gently pipetting five times using a P1000 pipette, resulting in a homogeneous cell suspension. The trypsin reaction was quenched by adding 200 μL of pre-warmed FBS (Sigma F0804) to the sample. Cells were centrifuged at $500 \times g$ for 5 min at 4 °C, and the supernatant was removed. Cells were resuspended in 1 mL of sterile-filtered ice-cold PBS supplemented with 2.5 μL of RNase inhibitors (Thermo Fisher Scientific N8080119). This process was repeated three times. The cell suspension was filtered through a 40 μm cell strainer (Falcon™ 352235), and the cell viability and concentration were assessed manually using Trypan Blue (Gibco™ 15250061), C-Chip disposable haemocytometer slide (NanoEnTek DHC-N01) on Leica DMLS light microscope (Leica 020-518.500). All samples had a final viability of at least 90%.

### scRNAseq library preparation and sequencing

Cell suspension concentration was adjusted to target 10.000 cells in each sample (i.e., developmental stage) and processed according to the Chromium Single-Cell 3' Reagent Kits User Guide (v3.1 Chemistry Dual Index). GEMs were obtained using the Chromium Next GEM Single-Cell 3' GEM Kit v3.1 (10x Genomics PN-1000130) and Gel Beads Kit v3.1 (10x Genomics PN-1000129). The scRNAseq library was generated using the Chromium Next GEM Single-Cell 3' Library Kit v3.1 (10x Genomics PN-1000196) and quantified using the Invitrogen™ Qubit™ dsDNA assay kits (FisherScientific 10616763) in a fluorospectrometer (ThermoFisher ND-3300). Sequencing was performed using the NovaSeq 6000 (PE100bp) platform, targeting a depth of 50,000 reads per cell, following 10X Genomics guidelines.

### In situ hybridization chain reaction

Hybridization chain reaction (HCR) followed the protocol for whole-mount samples (HCRv3 FISH) provided by Molecular Instruments, Inc., USA, with minor modifications. Briefly, fixed embryos were bleached with 10% peroxide in Dent's fix (90% Methanol, 20% DMSO) from 2h to 24h at room temperature and rehydrated in decreasing methanol steps 10 min each (75%, 50%, 25%, 0%–in 0.1% PBST) at 4 °C on slow rotation. Before dehydration, embryonic faces were dissected by cutting below the otic vesicle and removing the protruding midbrain (E9.5, E10.5), or below the mandible (E11.5–E14.5) and removing the brain, to obtain a flat head profile. Alternatively, the mandible of E13.5 and E14.5 embryos was also removed to expose the palatal shelves. Following dehydration, samples were post-fixed for 15 minutes at RT in 4% PFA, washed 3 × 5 minutes in PBST, and pre-hybridized in 30% probe hybridization buffer for 30 minutes at 37 °C. Afterwards, the samples were incubated from 16 h (E8.5–E11.5) to 48 h (E12.5–E14.5), depending on the tissue size, at 37 °C in a 2 pmol probe diluted in 30% probe hybridization buffer in a thermomixer (Eppendorf 5355) with gentle shaking. Samples were washed 4 × 15 min in 30% probe wash buffer at 37 °C, followed by 2 × 5 min washes with 0.1% 5× SSC-Tween (SSCT) at RT (20× SSC; Fisher bioreagents BP1325-4) and incubated in 30 pmol of fluorescent-labeled hairpins (488, 546, 594 and 647) diluted in amplification buffer at RT overnight, in the dark. After 4 × 15min washing in 5× SSCT, samples were incubated overnight at 4 °C in 1× DAPI in 5× SSCT to stain the nuclei. Finally, samples were washed

4 × 15 min in 0.1% PBST and dehydrated in progressively higher concentrations of methanol (25%, 50%, 75%, 100% in 0.1% PBST) for 10 min (E8.5–E11.5) or 30 min (E12.5–E14.5) each, before clearing with BABB (one part benzyl alcohol to two parts benzyl benzoate).

The embryonic facial samples were then mounted facing downwards into commercial glass-bottom dishes (VWR 734–2904) or an in-house-made metal slide with a glass-bottom well, embedded in BABB and covered with glass cover slips for imaging. The metal slide was used for bigger samples (E13.5 and E14.5) or scans longer than 6 hours.

Whole-mount samples were imaged on a Zeiss LSM980 Airyscan 2 inverted laser confocal microscope, using ×10 magnification. The tile scan was used for E11.5-E14.5 samples, set with a 10% overlap. The speed and acceleration of the stage were set to 10%. Tile scans were stitched using the channel DAPI as reference, single pixel correction was applied, and brightness and contrast were adjusted accordingly before making the 3D reconstructions or exporting the z-sections as TIFF images. All image processing was done using the Zen Blue v3.3 software provided by Zeiss. Raw microscopy images are deposited at https://evolomero.evolbio.mpg.de/gallery/show_dataset/1/.

Gene probe sets were either designed and purchased from Molecular Instruments, Inc. or designed using the HCR 3.0 probe maker (https://github.com/rwnull/insitu_probe_generator) and purchased from Integrated DNA Technologies (IDT, USA). Fluorescent-labeled hairpins were purchased from Molecular Instruments, Inc. Amplification and hybridization buffers were prepared in-house following the manufacturer's instructions[74]. Probes obtained from IDT are provided in the Suppl. Table 2.

## Single-cell RNA-seq data processing

In addition to the newly generated sequencing data, we included seven samples from the previously published La Manno et al. (2021) dataset spanning E8.5-E10.5. The 10x Genomics Cell Ranger (7.1.0) pipeline was used to process the RNA-seq data. First, a custom mouse reference was constructed based on the mm39 assembly using cellranger mkref. Reads from all samples were aligned to this reference and quantified using Cell Ranger Count. The raw count matrix was further processed in Python (3.9.16) using scanpy (1.9.3) (https://scverse.org/). Doublet detection was performed using Scrublet (https://github.com/swolock/scrublet). The threshold for predicted doublets for each sample was set based on the multiplet rate table provided by 10X Genomics, and predicted doublets were subsequently removed. Quality control metrics were calculated using pp.calculate_qc_metrics. Each sample was evaluated individually, and low-quality cells were removed based on these metrics (Supp. Information), resulting in 58,973 cells covering developmental stages E8.5-E14.5. Gene expression was normalized and log-transformed with pp.normalize_total and pp.log1p, respectively. Highly variable genes were identified with pp.highly_variable_genes. The cell cycle state of each cell was scored using tl.score_genes_cell_cycle based on the cell cycle markers identified in ref. 75. The individual data sets (from different developmental stages) were integrated using the topic modeling strategy of MIRA. The model was initiated with the individual samples as categorical covariates and the cell cycle score as continuous covariates. The model was trained on 10 topics. The topic compositions were transformed into Euclidean space using the predict function from MIRA for subsequent nearest-neighbor analysis in scanpy using pp.neighbors. Based on this nearest-neighbor graph, a UMAP embedding was constructed, and cells were clustered using tl.umap and tl.leiden, respectively. Clusters were annotated to major cell types based on known marker genes. Cells belonging to the mesenchymal and ectodermal populations were subset separately for further, higher-resolution analysis. Single-cell RNA-sequencing data generated in this study have been deposited at the Sequence Read Archive (SRA) and assigned the identifier PRJNA1243320.

## Analysis and annotation of mesenchymal cells per stage

For higher resolution clustering, annotation was performed stage by stage. When multiple samples were available for a given stage (i.e., E8.5 and E10.5), integration strategies were applied to negate batch effects: for the E8.5 samples we used the harmonypy algorithm (https://github.com/slowkow/harmonypy), and for the E10.5 samples, the scVI integration algorithm (https://scvi-tools.org/), as harmony was unable to resolve the double batch effect of our E10.5 sample (C57BL/6 N background) and E10.5 samples (CD1 background) from La Manno et al. (2021). As before, gene expression was normalized and log-transformed. The cell cycle was regressed from the cell cycle scores using pp.regress_out. A first dimensionality reduction was performed with tl.pca on the top highly variable genes identified with pp.highly_variable_genes. Further dimensionality reduction was performed with tl.umap and clustering with tl.leiden. Cells not relevant to our analysis (i.e., blood, immune cells, glia, endothelial cells, mesoderm, muscle, heart, brain, endoderm, eye, otic capsule) were removed based on marker genes. Individual clusters were annotated based on their gene expression, validated by HCR, and information from publicly available databases (i.e., Genepaint, Allen Brain Atlas).

The cranial neural crest and mesenchymal cells from the different stages were integrated using the MIRA Topic modeling strategy described above. The highly variable genes for the model were selected in two steps to prioritize transcription factors; first, pp.highly_variable_genes was run on a subset of genes known to be transcription factors (https://resources.aertslab.org/cistarget/tf_lists/), then pp.highly_variable_genes was run on the remaining genes with a cutoff of 500 genes. The topic model was initiated based on the list of TF-biased genes, with individual samples as categorical covariates and cell cycle score as continuous covariates. The model was trained with 24 topics. The topic compositions were transformed into Euclidean space using the predict function from MIRA for subsequent nearest-neighbor analysis in scanpy using pp.neighbors. A UMAP embedding was constructed using tl.umap with min_dist set to 0.01 and negative_sample_rate set to 4.

## Trajectory inference

The Velocyto command line tool (https://velocyto.org/) was used on the cellranger output to generate spliced and unspliced counts. These counts were added as layers to the anndata object. The scVelo package (https://github.com/theislab/scvelo) was used to perform RNA velocity analysis. Pseudotime analysis was performed using Palantir (https://github.com/dpeerlab/Palantir). A root state was selected based on stage (E8.5) and gene expression (CNCC markers). Terminal states were selected based on stage and gene expression. The Palantir algorithm was run on 10,000 waypoints for 100 iterations. Palantir pseudotime was used as the basis for the pseudotime kernel in Cellrank (https://github.com/theislab/cellrank). Cellrank is a modular framework for studying cellular dynamics based on Markov state modeling. The Cellrank GPCCA estimator was used to compute the top seven terminal states based on the pseudotime kernel. These terminal states reflected the presence of cells committed to a particular differentiation trajectory (e.g., osteogenic or chondrogenic) or comprising key anatomical structures (e.g., whisker pad or philtrum). Cellrank was used to derive trajectories for each of these end states. Based on these trajectories, fate probabilities and driver genes were calculated for each developmental lineage using the Cellrank pipeline. Driver gene expression trends were fitted to each lineage in pseudotime using the mgcv package (https://cran.r-project.org/web/packages/mgcv/index.html) in R and plotted using Cellrank.

## Integration of scRNAseq data with GWAS

To identify genes involved in the establishment of human facial morphology, we used the information from publicly available GWAS studies and reported clinical variants related to facial development.

GWAS variants associated with normal human facial shape variation (Supp. Data 4), as well as genes associated with pathologies not related to craniofacial development (Supp. Data 5), were obtained from the OpenTargets platform (version 24.06) and genetics (version 22.09), using relevant experimental factor ontology (EFO) classes and their descendants. Genes associated with the selected EFO classes with a score greater than 0.1 were retained for further analysis. Orthologous mouse genes were selected using the ENSEMBL human-to-mouse orthology map. Genes implicated in human craniofacial abnormalities were retrieved from DISGENET (v25.1.1, under the academic plan) (https://disgenet.com/), using terms C0158646, C0008924, C0008925, C0685787, C1850256, C1844537, C4759655, C0376634, C4023011, C2931196, C0221356, C0266617, C3665865 (Supp. Data 6). Similarly, the MGI Mammalian Phenotype Ontology Annotations database (https://www.informatics.jax.org/vocab/mp_ontology/) was queried for sets of genes associated with the mammalian phenotype term Abnormal Craniofacial Development (Supp. Data 7). To reduce background noise in all our gene sets, only those genes in each set that overlapped with our highly variable genes were retained. Duplicate genes for each search term were removed. The resulting gene lists were used as a gene set in conjunction with the single.pathway_aucell function from the omicverse package (https://github.com/Starlitnightly/omicverse). AUCell scores were calculated for each search term, which were then used to assess whether the gene set was overrepresented in specific cell types in our dataset. AUCell dot plots were created with matplotlib, and Z scores were calculated per stage to highlight differences in AUCell scores among clusters within a given stage. To visualize the distribution of AUCell scores, density curve plots were created using the kdeplot function in Seaborn (https://seaborn.pydata.org/index.html). The tl.rank_genes_groups function from scanpy was used to calculate the z-score and logfold changes for each gene in each cluster and at each stage. This Z-score was used as a cutoff ( > 4) to only show relevant logfold changes in the Supp. Data 4, 6, and 7. Localized Marker Detector (LMD; https://github.com/KlugerLab/LocalizedMarkerDetector) was used to calculate a marker score for genes in Supp. Data 4, 6, and 7 to highlight genes with localized expression. Genes associated with the cell cycle (at all stages) and batch effect (for the samples at E10.5) were removed before applying the LMD algorithm. Similarly, genes represented by fewer than 50 transcripts or present in >80% of cells were also excluded, as is advised in the LMD vignette. Empty slots in the LMD score column in Supp. Data 4, 6 and 7 are a result of these genes having been excluded from LMD calculations. The function KneeLocator from kneed (https://github.com/arvkevi/kneed) was used to calculate the knee point of the LMD scores for all cells and per stage. Genes with an LMD score below the knee point were marked in green in Supp. Data 4, 6 and 7 to highlight their relevance. In Supp. Data 4, human facial features and mouse clusters were matched manually and highlighted in yellow. Mouse clusters with logfold values >1 were marked in yellow and compared to the human feature as follows: human feature Nose was matched to mouse clusters LNP, nasal cavity, ectomesenchyme, chondrocytes, perichondrium, nasal palate and Roof-Sim1. Human nostrils to mouse clusters' nostrils, and lip/nostril fusion. Human Lip to mouse lip-dermal, dermal condensate and lip/nostril fusion. Human Philtrum to mouse philtrum. Human Cheek to mouse whisker pad. Human Whole Face Without Nose to palatal shelf, incisor, whisker pad. Spatially restricted mouse genes not matching an equivalent human facial feature were highlighted in orange. Human facial features not included in our study were marked in red (e.g., mandible).

## Prediction of cellular communication during development

To infer, analyze, and visualize cell–cell communication networks from single-cell RNA-sequencing data, we used CellChat V2 (https://www.cellchat.org). Cell-cell communication was analyzed separately at each stage, and analysis was restricted to the mesenchymal and ectodermal populations. Raw counts were normalized to 10,000 total counts per cell and log-transformed before running the CellChat pipeline. After preliminary analysis, we noticed an overrepresentation of the MK/PTN pathway family, which belongs to the secreted signaling communication class. To gain better insight into other secreted signaling molecules used by cells during development, we created a custom CellChatDB in which MK/PTN signaling was analyzed separately from the other secreted signaling pathways. The rest of the CellChat pipeline was followed using default parameters. The diagonal scatter plots were generated using the netAnalysis_signalingRole_scatter function in CellChat. The CellChat function netVisual_heatmap for generating heatmaps using the ComplexHeatmap package (https://github.com/jokergoo/ComplexHeatmap) was customized to include k-means clustering on both the rows and columns to highlight patterns in signaling, and signaling type annotations from the CellChatDB were added to each row to indicate which signaling type each pathway is annotated to.

### Reporting summary

Further information on research design is available in the Nature Portfolio Reporting Summary linked to this article.

## Data availability

The Single-cell RNA-sequencing data generated in this study have been deposited in the SRA under the accession code PRJNA1243320. These data are publicly available as of the date of publication. All presented scRNAseq data and microscopy images of gene expression mapping are provided in a user-friendly, interactive online database: https://www.evolbio.mpg.de/murillo-seton2025. This study obtained publicly available data from La Mano et al. (2021), accessible from the SRA under accession number PRJNA637987.

## Code availability

All of the Python and R packages used in this study are publicly available online. The list of packages and their versions, and the scripts used in this study for figures and analysis are available at: https://github.com/craniolab/Positional-programs-in-facial-variability/.

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

## Acknowledgements
The authors thank the Max Planck Society for the support. We are grateful to the animal facility staff at the Max Planck Institute for Evolutionary Biology and to Luiz Felipe Colli Sinhorini for their technical support, and NFDI4BIOIMAGE consortium for helpful advice. The sequencing service was provided by the Institute of Clinical Molecular Biology Kiel (IKMB-Kiel).

## Author contributions
A.M. performed sample collection, laboratory procedures, and data analysis and wrote the original draft. L.S. performed data analysis and edited the manuscript. E.E. conducted sample collection and laboratory procedures. A.D. performed data analysis. J.F. carried out sequencing. C.F.G. created the online atlas. M.K. designed and supervised the study, performed sample collection and data analysis, provided resources, wrote the original draft, and edited the manuscript. All authors read the manuscript and agree with the submitted version.

## Funding

## Competing interests
The authors declare no competing interests.
