## [Peer Review file · Nature Communications]

Positional programs in early murine facial development and their role in human facial shape variability

Corresponding Author: Dr Markéta Kaucká

Version 0:

Reviewer comments:

Reviewer #1

(Remarks to the Author)

The manuscript by Muillo-Rincon et al., describes an ambitious study designed to define positional programs during facial morphogenesis. The authors apply scRNA-seq to assess gene expression in specific facial growth centers and other cell populations involved in facial morphogenesis of mouse embryos from E10.5-14.5, and complement these new data sets with similar publicly available data sets from earlier mouse development (E8.5-10.5). As an important complement, the authors then assess spatial expression of selected genes across developmental time points and structures. These analyses provide extensive evidence of heterogeneity within early embryonic facial growth centers, including cranial neural crest-derived mesenchymal populations. The authors complement this main thrust of the work with analyses of ectodermal-mesenchymal signaling as one putative driver of the observed mesenchymal patterning. Finally, GWAS data is integrated into the described developmental trajectory in an attempt to link variants in specific genes to observed face shape characteristics.

The overall study is impressive in ambition and scope and the generated data will be of immediate and long-term value. While previous evidence has already established that the facial mesenchyme is not homogeneous, this work reveals new subpopulations and contextualizes these subpopulations across both space and time in a way that far supersedes any previous work. Many of the individual signaling pathways and gene expression domains represented in the manuscript have previously been characterized in individual publications, but the unification of these findings in a cohesive developmental atlas represents a major advancement that will surely be of tremendous value for future investigations into the mechanisms of craniofacial morphogenesis. The overall utility of this work to the field will depend in part on the characteristics of the interactive database of transcriptomic profiles and gene expression visualization that will be made available following publication. Recognizing that the online database is outside the scope of this review, it should be noted that the data present within the manuscript and supporting information is itself extraordinarily expansive and useful. These data will also be of likely value in understanding the role of individual genes and genetic variants in human face shape variation and dysmorphology, as suggested by the few examples provided by the authors.

As this work will serve as a powerful database for future studies, it is also important to assess its rigor. Comparison of a subset of signaling pathways and individual genes documented in this manuscript against previously published work supports the rigor and reproducibility of the data shown in the present manuscript. The clarity of expression domains visualized by HCR is far superior to the majority of previous documentation. And again, the comprehensive nature of this analysis serves to contextualize the findings of hundreds of individual analyses published over the last 20 years.

Overall, this work is innovative in scope, furthers our understanding of the positional programs during facial morphogenesis, and will serve as a high value resource for the field. The work may be additionally strengthened by considering the minor comments below.

1. Previous work has clearly demonstrated that the facial mesenchyme is not heterogeneous, and that patterning precedes differentiation. While the present work certainly provides additional resolution to that understanding, it is recommended to tone down the related claims of novelty.
2. The methodology description for multiplex HCR imaging should be expanded and clarified. For example, how were tissues imaged and how were digital sections created from wholemounts?
3. The analysis combines publicly available data from E8.5-10.5 with new data from E10.5-14.5. That the former was generated from CD-1 mice, while the latter was generated from C57BL/6N should be clearly stated. While not expected to

create incompatibility, this is an important variable to report.

Reviewer #2

(Remarks to the Author)

This is a very detailed and well written analysis of single cell data from mouse craniofacial tissues. I am impressed by the work and intrigued by the overall findings, but have some areas where the manuscript would benefit from further development to make it more well-rounded and impactful.

Can the authors say more about the set of genetic variants obtained from OpenTargets? And how many genes were annotated with the facial morphology measurement term? It would be helpful to understand which studies these came from, especially as more recent studies point to hundreds of loci with highly localized effects compared to earlier studies using larger measurements or distances that point to relatively few loci.

The authors end the results by stating that their work “This work also provides valuable insights into the effector cell populations contributing to congenital craniofacial disorders”, however, they did not systematically investigate any human gene lists related to craniofacial anomalies. This seems like an essential addition to the manuscript, even if one might predict the results to be similar to mouse gene list. There are many existing databases and recent publications that would make this feasible. Without such an analysis, these sort of statements should be removed from the results and limited to speculation in the discussion.

One of the strengths of the manuscript is the very detailed analysis of the single-cell data which speaks to a granularity of gene expression, cell type, and location in governing facial development. However, the AuCell analysis doesn't take advantage of this. Can the authors add more granularity to the facial morphology gwas results (which as mentioned above also point to very granular effects) or stratify the types of abnormal craniofacial development? As the authors do not clearly state the number of genes in the results or methods, it is difficult to determine if the issue statistical power? From a purely scientific sense, the authors seem to be missing an opportunity to generate insights into highly specific craniofacial anomalies or aspects of facial variation.

Minor comments:

Because the introduction ends with several sentences about humans, it would be helpful to add early in the results section (which precede the methods) that the face that is dissected is a mouse face. There was a time when this would seem obvious but there are now multiple similar studies using human fetal facial tissue and it is essential to now specify the species under study.

Similarly in lines 379-387 it would be appropriate to clarify in the text when the authors are referring to human phenotypes vs. mouse phenotypes.

Lines 119-123: These descriptions may be confusing to some readers, especially trainees; the authors might consider including a figure in the supplement to make it explicit what positions of the face are being designated by each term.

Reviewer #3

(Remarks to the Author)

This manuscript presents a single-cell atlas of mouse head and face development during early embryonic stages. The dataset enables readers to explore cellular expression dynamics from early cranial neural crest cells (CNCCs) to their terminal mesenchymal derivatives, spanning mouse embryonic days E8.5 to E14.5. The study serves as a valuable reference for molecular and developmental biologists interested in mouse craniofacial development. It may also aid in the development of new *in vitro* differentiation protocols, by revealing the cascade of positional gene expression programs required to drive cell-specific differentiation from CNCCs to terminal cell types relevant to head and face formation. The authors should be commended for their impressive multiplex in situ hybridization figures, which illustrate the expression of key positional marker genes in early mouse development.

Major feedback:

1. The manuscript presents the study of a mouse single-cell dataset, yet the authors' title and abstract suggest that human face shape variability would be explained. This is a stretch and borderline misleading. To make the connection between mouse single-cell data and human phenotype, the authors have compared genes important for human facial variation—both normal-range and abnormal—from several sources and have cross-referenced these genes with their scRNA-seq analysis. However, this exercise yields weak conclusions, some of which are already common knowledge. For example, it is neither novel nor surprising that many facial GWAS genes are expressed in mesenchymal cell populations which comprise the majority of the embryonic and, eventually, the postnatal face where morphometric measurements were conducted for GWASes.

2. The authors have merged datasets from two sources yet have not demonstrated that integration of the datasets could be accomplished successfully. In La Mano et al. (citation #18) the data are derived from CD1 mouse embryos, however in the current study the data are from C57BL/6 mouse embryos. What mouse strain-to-strain variability exists during development,

or what differences in tissue acquisition and library preparation among these two datasets could introduce batch effects that need to be corrected for? The authors should show the appropriate quality controls to verify that these datasets can be integrated. For example, one simple method is to show the UMAP plots before and after integration to validate that biologically identical cells are grouped together, rather than separated by batches.

3. The different cell populations at varied embryonic stages cannot be clearly discerned in the current study because all developmental stages are integrated together in all analyses. In Figure 1, for example, the UMAP plot can be recreated for each developmental stage (i.e. E8.5, E9.5, etc) to show the different cell populations present, and their relative abundance, at each timepoint. Figure 2F attempts to show the differences in the mesenchymal population by developmental stage but the differences are lost because different types of mesenchymal cells across all timepoints are overlapping on this plot.

4. Overall, I find that the manuscript focuses too heavily on raw analyses from software programs (CellChat, RNA Velocity, AUCCell, etc). Well-formulated hypotheses and interpretation of these raw software results within each result subsection are minimal. The manuscript should be revised to make the novel findings of this dataset clearer and more digestible for the reader.

Specific comments regarding main figures:

Figure 1: Regarding epithelial-to-mesenchymal transition (EMT), can the authors highlight genes in the dot plot of Figure 1B that demonstrate this process, such as *Cdh1*, *TWIST1*, and *Prrx1*, particularly during the E11.5 to E12.5 transition? In Figure 1A, the authors show a CNCC population outside of the ectodermal and mesenchymal clusters (i.e., the lime green cluster). Are these lime green CNCCs undergoing EMT? Along similar lines, how are the CNCCs within the mesenchymal population shown in Figure 1E different from the lime green CNCC population?

Figure 2: The integration of all development stages obscures the unique cell populations present in each. I would recommend the authors to present Figure 2A per developmental stage, rather than showing them in a combined plot such as with Figure 2F. In Figure 2F, it is particularly difficult to discern cell populations beyond E8.5.

Figure 4: Panels 4A, 4B, and 4C could be labeled or described more clearly in both the legend and the main text.

- For readers unfamiliar with CellChat, the authors should clearly state in the legend (and in the main text around line 285) that CellChat predicts four signaling categories, and should name these categories explicitly.
- The nomenclature used in CellChat also needs clarification. In Figures 4A and 4B, please define terms like “incoming” and “outgoing” signaling, and clarify what “probability” refers to. Figure 4A and B also appear cluttered with many terms (e.g., CADM, CDH, ADGRL) that are not discussed in the main text or figure legend. These could be removed or moved to supplemental figures.
- In Figure 4A and B, what I would be more interested in is the actual percentages of each of the orange, green, blue and purple bars. Please consider labeling them directly or summarizing in a supplemental table.
- In Figure 4C, please clarify the difference between interaction “strength” (on the axes) and interaction “count” (represented by bubble size).
- Finally, statistical tests and corresponding p-values should be included in the legend or main text to support the findings in Figures 4A, B, and C.

Figure 5: Please label the mesenchymal regions in Figures 5G–J (E12.5/E13.5) to be consistent with the epithelial regions labeled in Figures 5A–F. After all, the figure title suggests that morphogen expression shifts from ectodermal to mesenchymal populations between E11.5 and E12.5, but this shift is not clearly visualized in anatomically analogous regions across developmental stages.

Figure 6: Regarding the “targeted screen for genetic variants associated with facial shape variation” (line 343), the authors should comment whether *Cas21*, *Gata2*, *Lrriq1*, and *Pax3* are the only genes overlapping with facial GWAS significant SNPs? Or are these four genes examples that the authors wish to highlight. With similar reasoning, in Supplemental Table 1 the authors should provide a table of ‘facial shape’ GWAS, OpenTargets, or MGI genes (i.e. all face shape related databases consulted in the manuscript), the development timepoint(s) they are expressed within, and which UMAP cluster(s), or cell populations, they are associated with. This information would be highly useful for readers interested in downstream analysis of this dataset without starting from scratch.

Version 1:

Reviewer comments:

Reviewer #1

(Remarks to the Author)

The authors have adequately addressed my initial concerns and comments, while retaining the value and strengths of the initial manuscript. I have no other concerns or comments.

Reviewer #3

(Remarks to the Author)

The authors have satisfactorily addressed each of my suggestions in the revised manuscript. These include clearer descriptions in the results section that include more thoughtful conclusions of each analysis produced from individual software packages; clearer and corrected figures and their corresponding legends; and the incorporation of new supplementary material.

I especially appreciate the authors' transparency in adding new supplementary figures and tables. Specifically, I like that the authors have now provided cellular contributions at individual developmental stages (E8.5 – E14.5) in Supplementary Figure 4–6 and Supplementary Figure 11. Additionally, Supplementary Tables 4 and 7 now provide a list of normal-range and dysmorphic craniofacial genes investigated in this study, along with their cellular locales and expression profiles in this dataset.

Overall, the revised manuscript presents a single-cell dataset resource that is more accessible to craniofacial biology researchers, and the analysis presented herein offers a strong foundation for future research.

Response to the Reviewers

Reviewer #1 (Remarks to the Author):

Reviewer #1 (Remarks to the Author):

The manuscript by Muillo-Rincon et al., describes an ambitious study designed to define positional programs during facial morphogenesis. The authors apply scRNA-seq to assess gene expression in specific facial growth centers and other cell populations involved in facial morphogenesis of mouse embryos from E10.5-14.5, and complement these new data sets with similar publicly available data sets from earlier mouse development (E8.5-10.5). As an important complement, the authors then assess spatial expression of selected genes across developmental time points and structures. These analyses provide extensive evidence of heterogeneity within early embryonic facial growth centers, including cranial neural crest-derived mesenchymal populations. The authors complement this main thrust of the work with analyses of ectodermal-mesenchymal signaling as one putative driver of the observed mesenchymal patterning. Finally, GWAS data is integrated into the described developmental trajectory in an attempt to link variants in specific genes to observed face shape characteristics.

The overall study is impressive in ambition and scope and the generated data will be of immediate and long-term value. While previous evidence has already established that the facial mesenchyme is not homogeneous, this work reveals new subpopulations and contextualizes these subpopulations across both space and time in a way that far supersedes any previous work. Many of the individual signaling pathways and gene expression domains represented in the manuscript have previously been characterized in individual publications, but the unification of these findings in a cohesive developmental atlas represents a major advancement that will surely be of tremendous value for future investigations into the mechanisms of craniofacial morphogenesis. The overall utility of this work to the field will depend in part on the characteristics of the interactive database of transcriptomic profiles and gene expression visualization that will be made available following publication. Recognizing that the online database is outside the scope of this review, it should be noted that the data present within the manuscript and supporting information is itself extraordinarily expansive and useful. These data will also be of likely value in understanding the role of individual genes and genetic variants in human face shape variation and dysmorphology, as suggested by the few examples provided by the authors.

As this work will serve as a powerful database for future studies, it is also important to assess its rigor. Comparison of a subset of signaling pathways and individual genes documented in this manuscript against previously published work supports the rigor and reproducibility of the data shown in the present manuscript. The clarity of expression domains visualized by HCR is far superior to the majority of previous documentation. And again, the comprehensive nature of this analysis serves to contextualize the findings of hundreds of individual analyses published over the last 20 years.

We sincerely thank the Reviewer for their time and effort in assessing our manuscript and for their helpful comments. We also greatly appreciate the Reviewer's positive evaluation of our work and the recognition of the ambition, scope, and rigor of the study, as well as the emphasis on the potential long-term value of the datasets and integrative analyses we present.

The interactive online database contains the scRNA-seq data presented in this paper, as well as confocal microscopy images (including additional images not included in the paper) that were used to map gene expression. The scRNA-seq data is hosted on our own CellXGene instance (<https://github.com/chanzuckerberg/cellxgene>), which provides a user-friendly way of producing a variety of figures, subsetting cells of interest, and running additional analyses (*e.g.* DEG analysis, dot plots, and dual gene expression plots) via the 'Visualisation in Plugin' (https://github.com/interactivereport/cellxgene_VIP). The available data includes annotated integrated data for all major cell types, as well as for the subset containing annotated integrated ectoderm and the annotated subset containing the neural crest and mesenchymal populations, both integrated and per stage. The microscopy data is hosted on our own instance of OMERO (<https://www.openmicroscopy.org/omero/>), where all images are easily accessible. This allows researchers to browse through z-stacks, focus on selected channels (genes) and modify all channel settings. Microscopy data is also linked to scRNA-seq data in CellXGene, where relevant images can be viewed when selecting a gene.

We are committed to maintaining this database in the future. We intend to continuously add new data from our ongoing and future studies of craniofacial development, making it an increasingly valuable interactive resource. As such, we hope to contribute to the advancement of craniofacial research by providing an accessible, comprehensive, and interactive resource that will facilitate new discoveries and support a wide range of future investigations. Of course, all raw sequencing and confocal imaging data are also deposited in public repositories; links are indicated in the Data availability section.

Please note that the databases still undergo final modifications (*e.g.*, designing a landing page with instructions on how to navigate the interface) and will be fully ready for public access upon manuscript acceptance.

Overall, this work is innovative in scope, furthers our understanding of the positional programs during facial morphogenesis, and will serve as a high value resource for the field. The work may be additionally strengthened by considering the minor comments below.

1. Previous work has clearly demonstrated that the facial mesenchyme is not heterogeneous, and that patterning precedes differentiation. While the present work certainly provides additional resolution to that understanding, it is recommended to tone down the related claims of novelty.

In response to your comment and to acknowledge that the heterogeneity of the facial mesenchyme has been previously demonstrated, we have removed this claim in the abstract (Lines 23-24), where it was previously stated.

2. The methodology description for multiplex HCR imaging should be expanded and clarified. For example, how were tissues imaged and how were digital sections created from wholemounts?

We included a detailed description of how whole-mount sample imaging was performed in the methods (Lines 591-619).

3. The analysis combines publicly available data from E8.5-10.5 with new data from E10.5-14.5. That the former was generated from CD-1 mice, while the latter was generated from C57BL/6N should be clearly stated. While not expected to create incompatibility, this is an important variable to report.

Thank you for pointing this out! Indeed, this information must be clearly stated; therefore, we specified the genetic background of the individual samples in the Results (Lines 79-81) and Methods (Lines 649-654). Additionally, we provide evidence of correct integration of samples generated from both CD1 and C57/BL/6N strains and show that the observed clustering and cell heterogeneity are not caused by different genetic backgrounds. Please see Supp. Fig.1 M-N.

Reviewer #2 (Remarks to the Author):

This is a very detailed and well written analysis of single cell data from mouse craniofacial tissues. I am impressed by the work and intrigued by the overall findings, but have some areas where the manuscript would benefit from further development to make it more well-rounded and impactful.

We sincerely thank the Reviewer for their careful evaluation of our manuscript and for the encouraging feedback on the scope, detail, and clarity of our single-cell analysis. We very much appreciate the constructive suggestions that helped us to strengthen both the impact and interpretability of our study.

Can the authors say more about the set of genetic variants obtained from OpenTargets? And how many genes were annotated with the facial morphology measurement term? It would be helpful to understand which studies these came from, especially as more recent studies point to hundreds of loci with highly localized effects compared to earlier studies using larger measurements or distances that point to relatively few loci

We thank the Reviewer for this helpful comment! We absolutely agree that providing clearer information about the gene and variant lists and the studies they came from is needed. Additionally, (and based on your comment below on providing more granularity), we agree that providing more fine-grained, cluster-level expression data to the presented data will also strengthen the manuscript and allow other researchers to link genes/variants to specific cell populations, based on the expression levels and expression specificity. As such, researchers may more easily focus on genes or clusters of particular interest and select relevant data for their own studies, facilitating further research of the molecular basis of facial development and variation.

Accordingly, in the revised Supp. Table 4, we provide detailed information on the GWAS data, including an expanded list of GWAS genes and variants associated with normal human facial shape variation retrieved from OpenTargets (now 140 genes), the source study (studyID), the associated human facial feature, and the log-fold expression and p-values for each gene, presented for the integrated dataset, as well as per individual cluster and each developmental stage.

Additionally, we introduced the Localized Marker Detector (LMD), a score indicating how restricted the expression of a given gene is to a specific area or of UMAP or cell type. In this way, the readers can browse the table and extract the expression values of a gene of interest, per cluster and per stage, and have a measure of how spatially-localized the gene expression is. We also provide a new summary figure for visualization of these high-granularity (cluster- and stage-level) results in the new Supp. Fig. 21.

The authors end the results by stating that their work “This work also provides valuable insights into the effector cell populations contributing to congenital craniofacial disorders”, however, they did not systematically investigate any human gene lists related to craniofacial anomalies. This seems like an essential addition to the manuscript, even if one might predict the results to be similar to mouse gene list. There are many existing databases and recent publications that would make this feasible. Without such an analysis, these sort of statements should be removed from the results and limited to speculation in the discussion.

Thank you for this advice! We agree that the analysis of the “human craniofacial abnormalities” gene list was missing. Thus, we compiled a list of 89 genes reported to be causal or contributing to facial abnormalities in humans (please see the new Supp. Table 5), retrieved from DISGENET (For details, please see methods, lines 696-699). In the new table, we provide the same extent of information as for the GWAS genes associated with normal variation. Therefore, the expression values of these genes can also be explored at the cluster and stage levels. We also provide a summary figure (Supp. Fig. 25) for visualization of these high-granularity results.

One of the strengths of the manuscript is the very detailed analysis of the single-cell data which speaks to a granularity of gene expression, cell type, and location in governing facial development. However, the AuCell analysis doesn't take advantage of this. Can the authors add more granularity to the facial morphology gwas results (which as mentioned above also point to very granular effects) or stratify the types of abnormal craniofacial development? As the authors do not clearly state the number of genes in the results or methods, it is difficult to determine if the issue statistical power? From a purely scientific sense, the authors seem to be missing an opportunity to generate insights into highly specific craniofacial anomalies or aspects of facial variation.

Thank you for this constructive comment! As described in the previous answers, we appreciate the opportunity to introduce a higher resolution into the analyses and data presentation. Upon expanding gene and variant lists and introducing more detailed information in the Supp. Tables 4 and 5 and Supp. Fig. 21 and 25, we restructured the entire section “Cellular sources of natural human facial shape variability” (starting with Line 357). Additionally, in the text, we now specify the number of genes in the individual gene lists (“gene sets”) (Lines 373, 416, 420). All gene sets used for the AuCell analysis are now presented in the Supp. Table 6.

Minor comments:

Because the introduction ends with several sentences about humans, it would be helpful to add early in the results section (which precede the methods) that the face that is dissected is a mouse face. There was a time when this would seem obvious but there are now multiple similar studies using human fetal facial tissue and it is essential to now specify the species under study.

Thank you for this comment! We have added the information on where individual datasets come from (mouse/human) throughout the manuscript.

Similarly in lines 379-387 it would be appropriate to clarify in the text when the authors are referring to human phenotypes vs. mouse phenotypes.

We absolutely agree; therefore, we have clarified whether we refer to human or mouse phenotypes throughout the entire section “Cellular sources of natural human facial shape variability” (starting with Line 357).

Lines 119-123: These descriptions may be confusing to some readers, especially trainees; the authors might consider including a figure in the supplement to make it explicit what positions of the face are being designated by each term.

Indeed, these descriptions may be very confusing to some readers. In response to your suggestion, we provide a scheme (Suppl. Fig. 3) depicting the anatomical positions that were used to describe the cell clusters in the developing face.

Reviewer #3 (Remarks to the Author):

Major feedback:

1. The manuscript presents the study of a mouse single-cell dataset, yet the authors' title and abstract suggest that human face shape variability would be explained. This is a stretch and borderline misleading. To make the connection between mouse single-cell data and human phenotype, the authors have compared genes important for human facial variation—both normal-range and abnormal—from several sources and have cross-referenced these genes with their scRNA-seq analysis. However, this exercise yields weak conclusions, some of which are already common knowledge. For example, it is neither novel nor surprising that many facial GWAS genes are expressed in mesenchymal cell populations which comprise the majority of the embryonic and, eventually, the postnatal face where morphometric measurements were conducted for GWASes.

We would first like to sincerely thank the Reviewer for the time and effort spent assessing our manuscript and for providing such helpful feedback!

We appreciate the comments and, to avoid any potential misinterpretation, we have modified the title to “Positional programs in early murine facial development and their role in human facial shape variability”. Additionally, we also included this information in the abstract (Line 25).

We believe that using mouse data to explore the cellular sources of human facial shape variation is both a valid and a very informative approach. To support the validity of mouse to human data extrapolation (particularly the conservation of positional programs in the mesenchyme), we screened available human embryonic scRNAseq datasets, specifically Xu et al. (2023) and Yankee et al. (2023) (See Reviewer-only Figure 1 below). These figures compare matching developmental stages between mouse and human single-cell datasets.

Please note that the publicly available human datasets contain fewer cells, represent more heterogeneous tissues (likely the entire head), and, importantly, lack detailed information on data acquisition and processing. Using the available information, we queried our mouse positional genes and confirmed the existence of localized gene expression in humans (see attached Figure 1). We observe positional gene expression patterns in human datasets similar to those in mouse. For instance, there is a clear separation of the frontonasal from the maxillary mesenchyme – please see *Alx1/3*, *Pax7* and *Tfap2b* (frontonasal) versus *Dlx1/2* and *Lhx6/8* (maxillary) at CS12-14 and CS15-16 in human, compared to E10.5 and E11.5 UMAPs in mouse. Additionally, we also observed similar restricted expression patterns at later developmental stages (mouse E14.5 vs human CS20), particularly of genes associated with specific facial structures, for example *Lef1*, *Runx2*, *Meox2* and *Wnt5a*.

We provide this data for review purposes only. Due to the lower quality of the human data and missing details on processing (together with the lack of access to the raw data), we cannot incorporate these comparisons into our study.

To further support the validity of extrapolating mouse positional programs to humans, we compiled a list of 140 genes associated with normal human facial variation retrieved from the OpenTargets database (please find the detailed description in the Method section, starting with Line 688, and new Supp. Table 4). By exploring the enrichment of this gene set in our mouse scRNAseq data, we observed a correspondence between the human facial trait and the analogous mouse cell population/cell type (please see values marked in yellow in Supp. Table 4 and text lines 392-396). In the methods section (Lines 721-730), we explain how the human facial features and mouse clusters were matched. This information is also included in the legend of Supp. Table 4.

Extensive research demonstrates that human craniofacial phenotypes are successfully recapitulated in mouse models, further supporting the validity of mouse-to-human extrapolation. Therefore, we are confident that our methodology is robust and that our results contribute meaningfully to understanding the molecular and cellular bases of both normal and abnormal human facial variation.

We also note that, while it may seem widely recognized that GWAS genes relevant to face shaping are expressed in the mesenchyme, this conclusion is largely inferred from studies where craniofacial GWAS genes consistently map to enhancers active in the cranial neural crest and embryonic craniofacial tissues (e.g. White et al., 2020; Claes et al., 2018; Zhang et al., 2022; Prescott et al., 2015). To our knowledge, only one study has actually provided quantitative evidence of candidate genes contributing to normal human face variation being expressed in the embryonic facial mesenchyme (Zhang et al., 2022; Extended

Data Fig. 2C). All the studies and analyses rely on bulk sequencing data, offering only general tissue-level enrichment without precise cell-type resolution.

Similarly, to our knowledge, only one study has mapped genes associated with facial abnormalities (Siewert et al., 2023), but it is biased towards cleft lip/palate risk genes and focuses only on the ectodermal population of the lambda junction. To the best of our knowledge, our study represents the first systematic mapping of GWAS loci to a developmental stage- and cell-type-resolved single-cell dataset of the mouse embryonic face. We are convinced that this approach provides a valuable framework for interpreting GWAS results in a cell-type context. In the long term, this high-resolution information has the potential to accelerate functional studies and may ultimately contribute to improved therapeutic strategies for craniofacial disorders.

Additional references not shown in the manuscript:

Siewert, A., Reiz, B., Krug, C., Heggemann, J., Mangold, E., Dickten, H. and Ludwig, K. U. (2023). Analysis of candidate genes for cleft lip ± cleft palate using murine single-cell expression data. *Front. Cell Dev. Biol.* 11, 1091666.

Zhang, M., Wu, S., Du, S., Qian, W., Chen, J., Qiao, L., Yang, Y., Tan, J., Yuan, Z., Peng, Q., et al. (2022). Genetic variants underlying differences in facial morphology in East Asian and European populations. *Nat. Genet.* 54, 403–411.

Positional gene expression comparison between mouse and human face development (Part1)

Xu et al., 2023 - <https://doi.org/10.1038/s41556-023-01108-w>
Data retrieved from GSE157329
Human samples from CS 12, 13 and 14, equivalent to mouse E10.5
After filtering: 955 cells, mean counts of 13095
Sample identifiers: 'h0', 'ht7'

Images show similar expression patterns of important transcription factors and marker genes in mouse and human at comparable developmental stages.

Positional gene expression comparison between mouse and human (Part2)

Xu et al., 2023 - <https://doi.org/10.1038/s41556-023-01108-w>
 Data retrieved from GSE157329
 Human samples from CS15/16, equivalent to mouse E11.5
 After filtering: 4620 cells, mean counts of 11487
 Sample identifiers: 'h5', 'h9a', 'h9b'

Images show similar expression patterns of important transcription factors and marker genes in mouse and human at comparable developmental stages.

Positional gene expression comparison between mouse and human (Part3)

Yankee et al., 2023 - <https://doi.org/10.1038/s41556-023-01108-w>
 Data retrieved from GSE197513
 Human samples from CS 20, equivalent to mouse E15.5
 After filtering: 3050 cells, mean counts of 7434
 Sample identifiers: 'cs20_01', 'cs20_02'

Stage comparison between mouse and human based on:
 Xue et al., 2013 - <https://doi.org/10.1186/1471-2164-14-568>
 We don't present mouse E15.5 data in pur study, thus we compare human CS20 to mouse E14.5.

Review-only Figure 1. Positional gene expression in mouse and human embryonic craniofacial tissue. Comparisons between available equivalent human Carnegie stages (CS) and mouse embryonic day (E) are presented for E10.5 and CS12, 13 and 14 (Part 1), E11.5 and CS 15,16 (Part 2), and E14.5 was used for the comparison with CS20 (Part 3).

2. The authors have merged datasets from two sources yet have not demonstrated that integration of the datasets could be accomplished successfully. In La Mano et al. (citation #18) the data are derived from CD1 mouse embryos, however in the current study the data are from C57BL/6 mouse embryos. What mouse strain-to-strain variability exists during development, or what differences in tissue acquisition and library preparation among these two datasets could introduce batch effects that need to be corrected for? The authors should show the appropriate quality controls to verify that these datasets can be integrated. For example, one simple method is to show the UMAP plots before and after integration to validate that biologically identical cells are grouped together, rather than separated by batches.

We thank the reviewer very much for bringing up this important point! We clarified the strategies used to integrate stages with samples originating from different datasets and genetic backgrounds (La Manno vs our data). Please find this information in the methods section (Lines 649-654), as well as in the main text (Lines 79-81).

To demonstrate the proper integration of both data sets, we now provide in the new Supp. Fig. 1, UMAPs before (Supp. Fig. 1M) and after (Supp. Fig. 1N) integration. The comparison of these two UMAPs shows that cells from both data sets (CD1 and C57Bl/6N) group together based on transcriptome similarity and not batch. After integration, the distribution of cells in the UMAP (Supp. Fig. 1O) reflects the developmental stage progression (*i.e.* early stage at the center, later stages at the periphery) and not to the sample origin, providing further support for a successful integration.

3. The different cell populations at varied embryonic stages cannot be clearly discerned in the current study because all developmental stages are integrated together in all analyses. In Figure 1, for example, the UMAP plot can be recreated for each developmental stage (*i.e.* E8.5, E9.5, etc) to show the different cell populations present, and their relative abundance, at each timepoint. Figure 2F attempts to show the differences in the mesenchymal population by developmental stage but the differences are lost because different types of mesenchymal cells across all timepoints are overlapping on this plot.

We thank the Reviewer very much for this comment! We absolutely agree with the reviewer in that in Fig. 1 and Fig. 2A, it is not possible to distinguish the cellular composition of each individual stage.

The goal of Fig. 1, however, is to provide an overview of the major cell types found in the developing face. Whereas Fig. 2a should demonstrate the extensive molecular heterogeneity of the facial mesenchyme.

In response to your comment and to clearly display how the molecular composition of the developing face changes over time, we modified Supp. Fig. 4-6, and now provide UMAPs of individual stages E8.5-E11.5, clearly showing the cellular composition and cell population relative abundance at each embryonic

stage. UMAPs of individual stages from E12.5 to E14.5 are provided in the new Supp. Fig. 10 (Please see also the response to the comment on Fig. 2 below).

4. Overall, I find that the manuscript focuses too heavily on raw analyses from software programs (CellChat, RNA Velocity, AUCCell, etc). Well-formulated hypotheses and interpretation of these raw software results within each result subsection are minimal. The manuscript should be revised to make the novel findings of this dataset clearer and more digestible for the reader.

We appreciate this concern; therefore, to make our motivation, hypotheses and conclusions clearer, we introduced the statements throughout the Results section (for instance, please see the Lines 85-86, 91-92, 122-124, 286-287, 367-368, 390-392, 415-417, 420-421, 430-432. Please note that further interpretations of the results can be found in the Discussion.

Specific comments regarding main figures:

Figure 1: Regarding epithelial-to-mesenchymal transition (EMT), can the authors highlight genes in the dot plot of Figure 1B that demonstrate this process, such as *Cdh1*, *TWIST1*, and *Prrxl1*, particularly during the E11.5 to E12.5 transition? In Figure 1A, the authors show a CNCC population outside of the ectodermal and mesenchymal clusters (i.e., the lime green cluster). Are these lime green CNCCs undergoing EMT? Along similar lines, how are the CNCCs within the mesenchymal population shown in Figure 1E different from the lime green CNCC population?

In Fig. 1, the CNCCs cluster corresponds to the E8.5 samples, which primarily capture migratory CNCCs. To represent the full spectrum of CNCCs present in the dataset, we have updated the E8.5 stage dot plot in Supp. Fig. 4, and included canonical marker genes for both delaminating and migratory CNCCs.

The cells labelled as CNCCs in Fig. 1B and 1E are the same cells: in Fig. 1B, CNCCs are shown together with all the other cell types recovered from the developing face at the analyzed stages, whereas Fig. 1E represents a subset of Fig. 1B, showing only CNCCs and their mesenchymal derivatives. To clarify this, we revised Fig. 1 and the corresponding figure legend. Specifically, in Fig. 1B, 1C and 1E, the term “Neural crest” was replaced by “CNCCs”, and this abbreviation is now explained in the figure legend. Additionally, in Fig. 1B, we delineated the CNCCs and mesenchyme clusters together, and added arrows pointing to the respective subsets in Fig. 1C and 1E.

Please note that in the mouse model, CNCCs undergo EMT and delamination at E8.5, with their migration ceasing at E9.5 in the cranial region. Therefore, there is no EMTs or CNCCs present between E11.5 and E12.5. We think the reviewer may be referring to the reported role of EMT during facial prominence fusion (e.g. Losa et al., 2018; <https://doi.org/10.1242/dev.157628>), taking place between E11.5 and E12.5 in the mouse. In this particular case, molecular and morphological signatures of EMT were detected in epithelial cells at the lambda junction. As this process affects only epithelial cells and does not involve CNCCs or their derivatives, it is outside the scope of our present study.

Figure 2: The integration of all development stages obscures the unique cell populations present in each. I would recommend the authors to present Figure 2A per developmental stage, rather than showing them in a combined plot such as with Figure 2F. In Figure 2F, it is particularly difficult to discern cell populations beyond E8.5.

We would like to note that the purpose of Fig. 2F is to demonstrate the cellular contribution of each developmental stage to the integrated dataset. It is not intended to show the different mesenchymal populations over time. We believe the Reviewer may be referring to Fig. 2A, which displays all the mesenchymal cell populations described in the text. We agree that displaying the cellular composition stage by stage is an important missing component, and have therefore modified Supp. Fig. 4–6 (E8.5–E11.5) and created a new Supp. Fig. 11 (E12.5–E14.5) to provide single-stage UMAPs demonstrating the cellular composition at each stage. These new UMAPS will be available in the online resource as well.

Figure 4: Panels 4A, 4B, and 4C could be labeled or described more clearly in both the legend and the main text.

- For readers unfamiliar with CellChat, the authors should clearly state in the legend (and in the main text around line 285) that CellChat predicts four signaling categories, and should name these categories explicitly.
- The nomenclature used in CellChat also needs clarification. In Figures 4A and 4B, please define terms like “incoming” and “outgoing” signaling, and clarify what “probability” refers to. Figure 4A and B also appear cluttered with many terms (e.g., CADM, CDH, ADGRL) that are not discussed in the main text or figure legend. These could be removed or moved to supplemental figures.
- In Figure 4A and B, what I would be more interested in is the actual percentages of each of the orange, green, blue and purple bars. Please consider labeling them directly or summarizing in a supplemental table.
- In Figure 4C, please clarify the difference between interaction “strength” (on the axes) and interaction “count” (represented by bubble size).
- Finally, statistical tests and corresponding p-values should be included in the legend or main text to support the findings in Figures 4A, B, and C.

Thank you for the feedback on Figure 4. We agree that Fig. 4 may look somewhat crowded; however, we believe that the specific information on the relative contribution of main signaling molecules identified by CellChat and their dynamic changes over time is very informative and allows the readers to focus on individual pathways. Therefore, while we did not modify Fig. 4, in response to the Reviewer’s suggestion, we additionally present a new plot indicating the relative contribution (percentage) of each main signaling category (ECM, cell-cell contact and secreted signaling) to incoming and outgoing signaling per stage in the new Supp. Fig. 20.

In response to the individual points:

- Panels in Fig. 4A, 4B, and 4C are now labeled. An explanation of the terms used in the labels is included in the corresponding figure legend.

- CellChat signaling types are now listed and described in Lines 298-304.

- “incoming” and “outgoing” terms are explained in Lines 320-325

- An explanation of what “signaling probability” means is now included in the legend of Fig. 4 (Lines 970-972).

- An explanation of what “interaction strength” and “interaction count” (referred to as shape size in the figure legend) mean is now included in the legend of Fig. 4 (Lines 982-983, 985-986)

- In the new Supp. Table 1, we now provide the p-values for every interaction predicted by CellChat for our dataset (Pathways and interactions) and a list of all the interacting partners included in the CellChat dataset that were used to make the predictions (Pathway overview). Additionally, in the new Supp. Tables 2 and 3, we now provide the interaction probabilities per stage, used to create Fig. 4A and 4B, respectively, as requested by the reviewer. This information is also included in the Fig. 4 legend (Lines 989-991). We would like to note that only significant interactions (based on the presented p-values in Supp. Table 1) were used to create the figures and/or reported in the main text.

Figure 5: Please label the mesenchymal regions in Figures 5G–J (E12.5/E13.5) to be consistent with the epithelial regions labeled in Figures 5A–F. After all, the figure title suggests that morphogen expression shifts from ectodermal to mesenchymal populations between E11.5 and E12.5, but this shift is not clearly visualized in anatomically analogous regions across developmental stages.

Thank you for pointing out that Figures 5G–J have no labels marking the relevant populations - we have added the corresponding labels.

Figure 6: Regarding the “targeted screen for genetic variants associated with facial shape variation” (line 343), the authors should comment whether *Casz1*, *Gata2*, *Lrriq1*, and *Pax3* are the only genes overlapping with facial GWAS significant SNPs? Or are these four genes examples that the authors wish to highlight. With similar reasoning, in Supplemental Table 1 the authors should provide a table of ‘facial shape’ GWAS, OpenTargets, or MGI genes (i.e. all face shape related databases consulted in the manuscript), the development timepoint(s) they are expressed within, and which UMAP cluster(s), or cell populations, they are associated with. This information would be highly useful for readers interested in downstream analysis of this dataset without starting from scratch.

Thank you for this comment! In the revised manuscript, we have clarified that *Casz1*, *Gata2*, *Lrriq1*, and *Pax3* belong to the 140 GWAS genes associated with normal human facial shape variation that were retrieved from OpenTargets, and that they represent selected examples of highly spatially restricted genes (Lines 392-398).

Following your advice, we present the list of GWAS genes associated with normal human facial shape variation retrieved from OpenTargets (please see the new Supp. Table 4), which was used to produce Fig. 6B-D, as well as the source GWAS study (studyID), and the log-fold expression of these genes, per cluster and per stage. Additionally, we also introduced the Localized Marker Detector (LMD), a score indicating how restricted the expression of a given gene is to a specific area of UMAP (*i.e.* cell population). Similarly, in the new Supp. Table 7, we also provide detailed information about the abnormal mouse facial prominence development gene set (MGI genes) as requested. The corresponding AUCell enrichment analysis is now presented in the new Supp. Fig. 24. All gene sets used in this study are now present in Supp. Table 6. Additionally, following the advice from Reviewer 2, we compiled a list of 89 human genes reported to be causal or contributing to craniofacial abnormalities in humans, retrieved from the DISGENET database, and analyzed their enrichment across populations in our mouse scRNAseq data (please see the new Fig. 6E-G). The newly created human facial abnormality gene list is presented in the new Supp. Table 5, and contains detailed information such as the source study (pmid), LMD, and log-fold expression change for each gene per cluster and per stage. We additionally present new summary graphs for visualization of these results in new Suppl Fig. 25. In order to incorporate the new information presented in Supp. Tables 4 and 5 and Supp. Fig. 21 and 25, we rewrote and expanded the section “Cellular sources of natural human facial shape variability” (starting with Line 357).

Once again, we would like to cordially thank the three reviewers for their time and constructive feedback that helped us improve the manuscript and data presentation, highlight the novelty, and provide solid support for the findings and conclusions! We appreciate the opportunity to introduce the suggested changes, and we hope the reviewers will find them satisfactory.